



# Paleoseismic history of the intermountain Rieti Basin (Central Apennines, Italy)

Franz A. Livio[1], Anna M. Blumetti[2], Valerio Comerci[2], Maria F. Ferrario[1], Gilberto Binda[4], Marco Caciagli[3], Michela Colombo[1], Pio Di Manna[2], Fernando Ferri[2], Fiorenzo Fumanti[2], Roberto Gambillara[1], Maurizio Guerra[2], Luca Guerrieri[2], Paolo Lorenzoni[5], Valerio Materni[6], Francesco Miscione[2], Rosa Nappi[7], Rosella Nave[7], Kathleen Nicoll[8], Alba Peiro[9], Marco Pizza[1], Roberto Pompili[2], Luca M. Puzzilli[2], Mauro Roma[2], Aurora Rossi[1], Valerio Ruscito[2], Vincenzo Sapia[6], Argelia Silva Fragoso[1], Emanuele Scaramuzzo[1], Frank Thomas[1], Giorgio Tringali[1], Stefano Urbini[6], Andrea Zerboni[10], Alessandro M. Michetti[1,7]

[1]Università degli Studi dell'Insubria, Como, Italy
[2]ISPRA, Geological Survey of Italy, Roma, Italy
[3]Istituto Nazionale di Geofisica e Vulcanologia, Sezione di Bologna, Italy
[4]Norwegian Institute for Water Research, Oslo, Norway
[5]Professional Soil Science Consultant, Rieti, Italy
[6] Istituto Nazionale di Geofisica e Vulcanologia, Sezione di Roma 2, Italy
[7] Istituto Nazionale di Geofisica e Vulcanologia, Osservatorio Vesuviano, Napoli, Italy
[8]University of Utah, Salt Lake City, Utah USA
[9]University of Zaragoza, Spain
[10]Università degli Studi di Milano, Italy

*Correspondence to*: F.A. Livio (franz.livio@uninsubria.it)

**Abstract.** From the paleoseismological and seismotectonic point of view, the intermountain basins of the Central Apennines of Italy are one of the most studied areas worldwide. Within this context, however, the Rieti Basin, bounded at its sides by active faults and with its peculiar rhombohedral shape, is a relatively overlooked area, and its most recent paleoseismological studies date back to the '90s. This is a key area both for completing the paleoseismological history of this sector of the chain and for understanding how the present-day extensional regime is accommodated, through time by the faults bounding the basin. With this aim in mind, we excavated 17 paleoseismological trenches along the normal faults bordering the Rieti Basin (Central Apennines, Italy) and unveiled 15 paleoearthquakes that ruptured the faults during the last ca. 20 kyr.

Our analysis of the paleoearthquake succession along the basin-bounding faults suggests that a temporal clustering of rupturing events characterizes the basin with a maximum credible earthquake of Mw 6.5, consistently within this sector of the Central Apennines. These results suggest that for the Rieti Basin, stress transfer among surrounding faults can be ascribed as one of the processes behind the temporal clustering of earthquakes.



## 1 Introduction

The Central Apennines in Italy are among the best studied areas worldwide in terms of active tectonics and paleoseismology
(Michetti et al., 1996; Pantosti et al., 1996; Galadini and Galli, 1999; Salvi et al., 2003; Roberts and Michetti, 2004; Galli et
al., 2008, 2011; Gori et al., 2015; Moro et al., 2016; Blumetti et al., 2017; Di Domenica and Pizzi, 2017; Cinti et al., 2018;
Iezzi et al., 2019, 2023; Roberts et al., 2025). Tens of paleoseismological trenches have been excavated along the active
normal faults of this mountain belt, providing one of the world's most complete datasets for an extensional domain, together
with the well-known Basin and Range Province of the western U.S. (McCalpin, 2009). This led to choosing the Italian
Central Apennines as one important global case study for the development of codes for fault-based seismic hazard
assessment (Faure Walker et al., 2021; Scotti et al., 2021).

Nonetheless, some overlooked areas within the Central Apennines lack comprehensive studies and investigations. This is the
case of the Rieti Basin (Fig. 1), an intermountain fault-bounded depression located in the axial sector of the range which,
following the contractional events that built the chain, experienced several phases of extensional tectonics during the Plio-
Quaternary, leading to the present day-configuration. Today, the Rieti Basin morphology appears as a box-shaped alluvial
plain, bounded by normal faults striking almost orthogonally to each other (Fig. 1). The master fault of the basin ("Rieti
Fault"; Roberts and Michetti, 2004, and references therein) is the ca. 21 km long Eastern Border Fault (Fig. 1). The Rieti
Fault is part of a belt of active Quaternary normal faults (including the nearby Fiamignano and Fucino Faults; Bosi, 1975;
Michetti et al., 1996; Mildon et al., 2022) that characterizes the western side of the Central Apennines (Cowie and Roberts,
50 2001).

Past research focused on the Plio-Quaternary tectono-sedimentary evolution of the basin (Cavinato, 1993; Calderini et al.,
1998; Cavinato et al., 2000, 2002; Guerrieri et al., 2006), but only a few studies from the mid-90s, revealed Late Pleistocene
to Holocene strong seismic events by means of a paleoseismological approach. Observed Holocene slip rates at Piedicolle
and La Casetta Sites are in the order of 0.2 - 0.4 mm/yr. This is in good agreement with slip-rates measured along post-
glacial bedrock fault scarps (Cowie and Roberts, 2001), and long-term slip rates estimated from displacement of Early
Pleistocene paleo-surfaces (e.g., Brunamonte et al., 1993; Michetti et al., 1995). In fact, some open questions are still
pending on the area, including: i) the real seismotectonic potential of the Rieti Basin faults; ii) the coexistence and
contemporary activity of two sets of ca. orthogonal normal faults under the same extensional domain; and iii) the possible
occurrence of earthquakes rupturing the whole length of the Rieti Fault.

In 2020-2022, the Italian Government Extraordinary Commissioner for the 2016 Central Italy post-earthquake reconstruction
funded a program of studies on faults affecting urbanized zones in the earthquake epicentral areas. Some of the inhabited
centers under study are located along the northern, eastern and southern borders of the Rieti intermountain basin; boxes in
Figure 1C show the selected Study Areas. In 2021 and 2022, our team discovered evidence of Late Pleistocene and Holocene
activity along the investigated faults from study of 17 new paleoseismic trenches with 52 radiometric dates, and extensive
geophysical prospecting. We describe trench siting and investigations on 3 sides of the basin, identifying several





paleoearthquakes that occurred in the last about 20 kyr. Based on these data, the paper aims at providing a summary of the paleoseismic history of the Rieti basin; we discuss Late Quaternary fault slip-rates, event chronology, and we present a space-time paleoseismological diagram to infer the possible sequence of single vs multi-rupture events that affected the basin.

## 2 Geological and seismotectonic setting.

The Rieti Basin is within the Central Apennines, a former Meso-Cenozoic passive margin that evolved in a fold and thrust belt during the Neogene westward subduction of the Adriatic plate. The present-day setting is the result of an eastward progressive migration of the front of the accretionary wedge and of the back-arc extensional domain (Doglioni, 1991; Cavinato and De Celles, 1999; Cosentino et al., 2010), with the axial sector of the mountain chain currently affected by strong normal fault earthquakes due to ongoing crustal extension (Fig. 1b).

The local paleogeographic domain is the Umbria-Marche Domain: a stack of tectonic units composed of Jurassic-Paleogene carbonate successions (i.e., the Mt. Sabini and Mt. Reatini units; Fig. 1c). The thrust sheets superposed on an inherited tectonic setting developed during Jurassic-Eocene rifting of the passive margin (Galluzzo and Santantonio, 2002; Santantonio and Carminati, 2011; Capotorti and Muraro, 2024), which was characterized by mainly N-S striking normal faults but locally represented by a dense grid of faults, also including ca. E-W and NW-SE normal faults (Capotorti and Muraro, 2024).

Presently, the Rieti Basin is a box-shaped morphological depression, bounded along all its sides by Quaternary normal faults. The development and evolution of the basin is closely related to the extensional regime that affected the western flank of the eastward migrating Apennine chain during the Quaternary (Calderini et al., 1998; Cavinato and De Celles, 1999; Cavinato et al., 2000, 2002; Roberts and Michetti, 2004; Guerrieri et al., 2006), in combination with regional uplifting (Dramis, 1992; Galadini et al., 2003). During the Early Pleistocene, the subsidence of the basin was controlled by the eastern border normal fault, striking ca. NNW-SSE (Fig. 1c). This structure, connected to a small monogenic volcano formed near Cupaello (Lustrino et al., 2025), acted as the master-fault, controlling a progressive deepening of a half-graben and promoting the deposition of thick alluvial fan deposits (i.e., the Fosso Canalicchio Synthem) and alluvial plain sequences (i.e., the Monteleone Sabino Synthem), pinching out to the west. During the Middle Pleistocene, along the margins of the basin, the Early Pleistocene deposits were displaced by a few hundred meters, with normal faulting slip rates in the order of 0.2 – 0.4 mm/yr (Michetti et al., 1995), with further deepening of the Rieti Basin. Regional uplift triggered strong fluvial erosion from the Nera River, which captured the Velino River through the Le Marmore threshold, and generated the northward diversion of the whole hydrographic network (Guerrieri et al., 2006). Later tectonic events developed other sets of faults, some of them in the inner portions of the basin, finally resulting in the orthogonal fault architecture presently bordering this intra-mountain depression. Faults also controlled fluid flow and, together with climate oscillations, the deposition and erosion of thick sequences of travertine, coincident with spring emergence and water flow mainly along the northern and southern border of



the basin (Carrara et al., 1992). The fast Holocene growth (ca. 160 m of vertical accretion) of the Le Marmore travertine platform dammed the Velino River and caused alluvial overfilling in the Rieti Plain with sedimentation rates of ca. 3 mm/yr

(Guerrieri et al., 2006; Archer et al., 2019). Fluvial erosion and deposition processes are therefore one order of magnitude more efficient than tectonic faulting in shaping the local landscape of the Rieti Basin.

As already mentioned, we focus on the three areas along the border faults as requested by the Government Commissioner. To the east, we investigated the master fault in the Cantalice Study Area; to the north, we investigated the N border fault at the Rivodutri Study Area; to the south, we studied the eastern sector of the S border fault, at the Rieti-Santa Rufina Study

Area (i.e., east of Rieti town; Fig. 1c). The extensive investigations conducted under the 2016 post-seismic reconstruction project provide a unique opportunity for a comprehensive characterization of local paleoearthquake surface faulting.

The seismicity of the Rieti Basin shows two ancient destructive events. The oldest (Io = X MCS; Me = 6.4) occurred in 76 BCE, of which, however, information is only available for the city of Rieti (Guidoboni et al. 2018, 2019). The second strongest event recorded here is the Dec. 1$^{st}$, 1298, earthquake (Mw 6.26; Io = X MCS; Brunamonte et al., 1993; Rovida et

al., 2022) whose epicenter is tentatively located to the north of the basin. Of note, a foreshock event occurred only a day before in the southern part of the basin (Mw = 4.4; Io = V-VI MCS).

Relevant M>6 earthquakes, with epicenters within 40 km from the Rieti Basin, are likely responsible for coseismic environmental effects recorded in the Rieti Basin itself (Archer et al., 2019). These include events that occurred on Sept. 9$^{th}$, 1349 (Valle del Salto, Fiamignano Fault, Me = 6.1; Bosi, 1975; Mildon et al., 2022), Oct. 15$^{th}$, 1639 (Amatrice, Laga Fault,

Me = 6.2; Galli et al., 2017; Mildon et al., 2017) and Jan 14$^{th}$, 1703 (Norcia Fault, Me = 6.7; Blumetti, 1995; Guidoboni et al. 2018, 2019; Galli et al., 2021).

Several moderate historical earthquakes hit the area during the last 3 centuries (Fig. 1 c; Rovida et al., 2022). The Oct. 9$^{th}$, 1785 (Mw 5.76; Io VIII-IX MCS) hit to the northwest of the area. Two relatively recent moderate earthquakes occurred near the N and S border faults, which we investigated with exploratory trenches (Fig. 1c). The one to the south is the June 27$^{th}$,

1898, Santa Rufina event (Mw 5.5; Io VIII-IX MCS; Comerci et al., 2003) whose causative fault, based on the epicentral area, could be either the eastern boundary fault or the southern boundary fault of the basin. A similar size earthquake hit the NE border of the basin near Rivodutri on December 31$^{st}$, 1948 (Mw 5.3; Io VIII MCS; Bernardini et al., 2013).





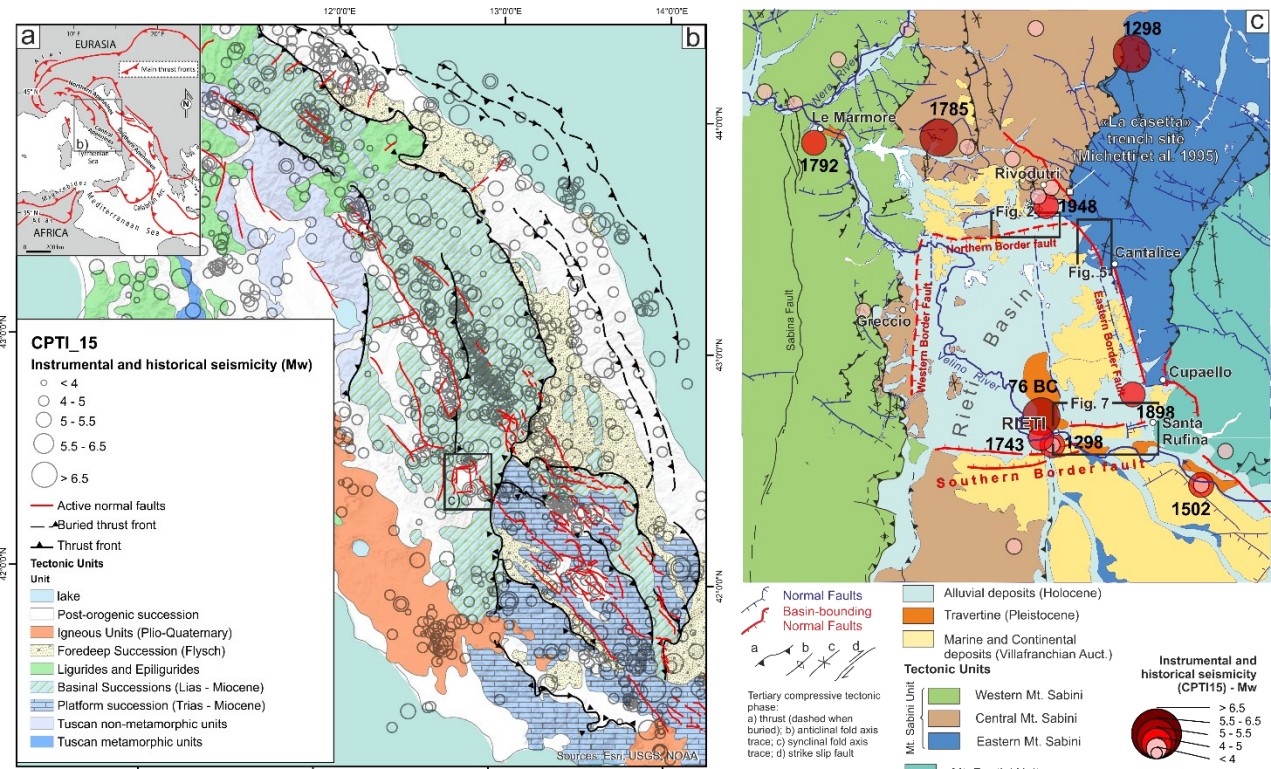

**Figure 1: Geological setting: a) regional tectonic framework; b) simplified geological map of the Central Apennines with thrust**
**fronts, the active normal faults running along the axis of the chain and seismicity (CPTI15 Catalogue, Rovida et al., 2022;**
**CFTI5Med Catalogue, Guidoboni et al. 2018, 2019; and Baranello et al., 2024, for the 1743 Rieti earthquake); c) Rieti Basin area:**
**simplified seismotectonic map (modified after Capotorti and Muraro, 2024; Servizio Geologico d'Italia, in press), the three Study**
**Areas are delimited by boxes; note in the NW corner of the Figure, the Le Marmore travertine platform dams the Velino River**
**Valley just before its confluence with the Nera River Valley, and therefore controls the drainage of the whole Rieti Plain.**

## 3 Materials and Methods


We investigated the three Study Areas by means of a multidisciplinary approach that aimed for best imaging the border faults and locating the paleoseismological trenches. For the purposes of data description, in the following we divide the three investigated Areas (namely along the northern, eastern and southern Border Faults) into Sectors, each one containing multiple Sites (i.e., individual trenches).

Geophysical prospecting included Electric Resistivity Tomography (ERT) and Ground Penetrating Radar (GPR) imaging. We acquired ERT and GPR lines across the main strands of the border faults.

ERT data was acquired using alternatively i) the X612EM+ multichannel geo-resistivimeter (MAE - Molisana Apparecchiature Elettroniche s.r.l.) capable of handling 96 electrodes connected to 4 bidirectional 24-takeouts cables; ii) a Syscal R2 multichannel geo-resistivimeter (IRIS Instrument) capable of handling 64 electrodes connect to 4 cables (16-

takeouts). Regarding the investigation strategy, measurements were taken by using at least two out of three different arrays



(i.e., among Wenner, Wenner-Schlumberger, dipole-dipole and pole-dipole) along each profile, exploiting the different sensitivity of each to achieve good resolution in both vertical and horizontal directions (see e.g. Dahlin & Zhou 2004). The sequences used during the surveys always consisted of collecting a large number of measurements, as it was preferred to adopt a "high-resolution" approach with an overabundant number of measurements for each layer/pseudo-depth. In particular, we used dipole-dipole and/or pole-dipole arrays almost at each Site, as considered most suitable for characterizing areas where both lateral and vertical variations in resistivity can be expected. This choice also made it possible to combine requirements for horizontal measurement coverage, resolution, depth of investigation as well as needs related to timing and logistics (e.g., temporary access to private areas). We carried out the 2D data inversion process after the quality check and data filtering using the Res2DInv software by Geotomo (Loke & Barker 1996). For each profile, each dataset derived from single array was processed individually and then, integrated into a single inversion process along with the others, to improve the quality of the final 2D model that would be adopted and interpreted.

We acquired GPR lines with a GSSI Sir4000 instrument equipped by a 200 MHz antenna and coupled with a differential GPS receiver Topcon GB1000 (post-processing kinematics) registering 1 position/sec. The acquisition parameters are summarized as follows: range: 150 ns; samples: 512; acquisition rate: 40 scan/s; dynamic: 32 bit. Data processing included vertical and horizontal filtering, gain levelling, deconvolution and migration. We converted Two Way Traveltimes (TWT) to depth using an average of dielectric constants calculated from diffraction hyperbolas occurring in the dataset and verified after the trench excavations.

We logged the paleoseismological trenches at the 1:20 scale, using a reference net, and by means of digital photogrammetry. For the latter, we used both portable LiDAR scanning (3D Scanner App for iOS) and a Structure from Motion and Multi-View Stereo workflow (SfM-MvS – Metashape software; Westoby et al., 2012; Bemis et al., 2014). Field descriptions and horizon designations followed internationally accepted guidelines (FAO, 2006), with the color definitions using nomenclature of the Munsell® System (1994). We also carefully considered the sediments, horizons, boundaries and the crosscut relations with structural features to chronologically constrain the progressive deformation affecting the stratigraphic sequences.

In the three Study Areas, we excavated 17 trenches, 13 of them suitable for paleoseismological analyses. We collected 52 samples for AMS dating, 21 from Rivodutri Municipality, 14 from Cantalice, 8 from Rieti and 9 from Santa Rufina (Cittaducale). We mapped the position of the samples on the trench logs, while Table 1 shows a summary of all the dated samples. The samples were analyzed by specialist laboratories (Beta Analytic and CEDAD); calibration was performed with the INTCAL20 curve using the OxCal ver 3.10 or the BetaCal4.20 software (CEDAD and Beta samples, respectively).

We also took into account results from the Michetti et al. (1995) trenching at 3 sites (Piedicolle, La Casetta and Caporio). Geochron Lab performed the radiocarbon dating of samples from these trenches; we revisited the original dates using the same calibration methods as above. Please note that throughout the text below we always refer to the calibrated ages listed in the last column of Table 1.



In the following text, we present results from individual trenches and ERT/GPR profiles that are most relevant for the paleoseismological analysis. Please refer to the Supplementary Material, which includes information on all trenches and geophysical prospecting realized during this project.

## 4 Results

We list the dated samples from the 13 paleoseismological trenches in Table 1, supplemented with the samples collected by

Michetti et al. (1995) in the La Casetta, Piedicolle and Caporio Sites. The Table presents both uncalibrated determinations and calibrated age. In order to keep the discussion consistent and understandable, when the probability distribution of these calibrated ages shows multiple peaks, in the following we will refer to the central age with the largest probability only (i.e., the one age calibration with highest probability; underlined in Table 1).

**Table 1: Summary of AMS dating results; we indicate the municipality and trench number, together with uncalibrated and calibrated ages; "Dated Material" column abbreviations: "Org" for organic materials, "Charc" for charcoal, or burnt plant materials; LTL samples are from CEDAD, CEntro di DAtazione e Diagnostica, Università del Salento, Lecce; Beta samples are from BETA Analytic, Inc., Miami.; GX samples (marked with * ) are C14 dating after Michetti et al. (1995), recalibrated using the Oxcal software and IntCal 20 curve; analyses from Geochron Lab., Cambridge, Massachusetts.**

| Municipality | Trench | ID sample | ID lab | Dated material | Uncal age (years BP) | Calibrated age (years) using IntCal20 curve |
|---|---|---|---|---|---|---|
| Rivodutri | APO T2 | C01 | LTL21211 | Org | 223 ± 45 | 1522-1574 CE (5.9%)<br>1625-1698 CE (29.0%)<br>1722-1814 CE (38.4%)<br>1835-1882 CE (3.8%)<br>1910 CE (18.3%) |
| | | C03 | LTL21259 | Org | 10947 ± 100 | 11127-10795 BCE (95.4%) |
| | | C04 | LTL21213 | Org | 6670 ± 45 | 5664-5512 BCE (90.4%)<br>5505-5481 BCE (5.0%) |
| | | C06 | LTL21214 | Org | 4404 ± 45 | 3327-3225 BCE (14.9%)<br>3183-3154 BCE (3.0%)<br>3110-2907 BCE (77.5%) |
| | | C07 | LTL21215 | Org | 4868 ± 45 | 3767-3721 BCE (6.6%)<br>3715-3603 BCE (69.5%)<br>3588-3528 BCE (19.3%) |
| | | C08 | LTL21216 | Org | 4955 ± 45 | 3914-3876 BCE (5.8%)<br>3804-3641 BCE (89.6%) |



| | | C09 | LTL21217 | Org | 4700 ± 45 | 3625-3560 BCE (17.3%) <br> 3533-3370 BCE (78.1%) |
|---|---|---|---|---|---|---|
| | | C11 | LTL21218 | Plant remains | After 1950 CE | After 1950 CE |
| Rivodutri | CAM | C04 | LTL22569 | Org | 15765 ± 120 | 17415-16872 BCE (95.4%) |
| | | C06 | LTL22570 | Org | 26571 ± 100 | 29147-28749 BCE (87.0%) <br> 28666-28486 BCE (8.4%) |
| | | C08 | LTL22571 | Org | 17734 ± 65 | 19879-19280 BCE (95.4%) |
| | | C10 | LTL22572 | Org | 18557 ± 75 | 20739-20353 BCE (95.4%) |
| Cantalice | CUC | C02 | Beta 634825 | Charc | 2280 ± 30 | 401-351 BCE (51.3%) <br> 302-208 BCE (44.1%) |
| | | C03 | Beta 634826 | Charc | 2440 ± 30 | 591-408 BCE (62.3%) <br> 751-684 BCE (22.3%) <br> 668-634 BCE (9.7%) <br> 622-613 BCE (1.1%) |
| | CANT-T1 | C01 | LTL21209 | Charc | After 1950 CE | - |
| | | C03 | LTL21210 | Charc | After 1950 CE | - |
| | | C05 | LTL21205 | Charc | 138 ± 40 | 1797 - 1945 CE (57.3%) <br> 1670 - 1780 CE (38.1%) |
| | CANT-T3 | C01 | LTL22573 | Org | 17860 ± 65 | 20000 – 19475 BCE (95.4%) |
| | | C02 | LTL22574 | Org | 13860 ± 65 | 15083 – 14651 BCE (95.4%) |
| | | C03 | LTL22575 | Org | 9797 ± 45 | 9326 – 9206 BCE (95.4%) |
| Rieti | TR 1 | C01 | Beta 631859 | Org | 8710 ± 30 | 7818–7597 BCE (94.6%) <br> 7931-7922 BCE (0.8%) |
| | | C02 | Beta 631860 | Orgt | 5000 ± 30 | 3811–3701 BCE (65.2%) <br> 3942-3865 BCE (24.2%) <br> 3682-3655 BCE (5.9%) |
| | | C03 | Beta 631861 | Org | 4310 ± 30 | 3011–2885 BCE (95.4%) |



| | | C04 | Beta 631856 | Org | 7510 ± 30 | 6438–6340 BCE (71.9%) 6313-6256 BCE (23.5%) |
| | | C05 | Beta 631857 | Org | 18670 ± 60 | 20919–20458 BCE (95.4%) |
| Rieti | TR 5 | C03 | Beta 631858 | Org | 4060 ± 30 | 2673–2474 BCE (88.7%) 2843-2813 BCE (6.3%) 2738-2735 BCE (0.4%) |
| Rieti | Villa Stoli South | C12 | CEDAD LTL21261 | Org | 19127 ± 75 | 21261-20961 BCE (95.4%) |
| | | C16 | CEDAD LTL21260 | Org | 11801 ± 55 | 11835-11630 BCE (85.0%) 11601-11563 BCE (10.4%) |
| Santa Rufina (Cittaducale) | TR 3 | C01 | Beta 632175 | Org | 5270 ± 30 | 4171-4036 BCE (60.9%) 4233-4192 BCE (18.8%) 4027-3987 BCE (15.7%) |
| | | C02 | Beta 632176 | Org | 8640 ± 30 | 7729-7589 BCE (95.4%) |
| | | C03 | Beta 632177 | Org | 4870 ± 30 | 3711-3627 BCE (87.7%) 3560-3534 BCE (7.7%) |
| | | C04 | Beta 632178 | Org | 4550 ± 30 | 3243-3102 BCE (57.2%) 3371-3306 BCE (35.1%) 3300-3283 BCE (2.1%) 3276-3266 BCE (1.0%) |
| | | C05 | Beta 632179 | Org | 5910 ± 30 | 4846-4712 BCE (95%) 4878-4875 BCE (0.4%) |
| | | C06 | Beta 632180 | Org | 6000 ± 30 | 4988-4797 BCE (95.4%) |
| | | C07 | Beta 632181 | Org | 1690 ± 30 | 326-424 CE (78.6%) 255-286 CE (16.8%) |
| | | C08 | Beta634827 | Char | 8700 ± 30 | 7799-7597 BCE (95.4%) |
| Santa Rufina (Cittaducale) | TR 4 | C01 | Beta 632182 | Org | 1770 ± 30 | 224-376 CE (95.4%) |
| Rivodutri, La | CAS | C01* | GX-16336 | Org | 1530 ± 80 | 382-657 CE (95.4%) |





| | | | | | | |
|---|---|---|---|---|---|---|
| Casetta | | C02* | GX-16338 | Org | 1635 ± 85 | 245-594 CE (95.4%) |
| | | C03* | GX-16335 | Org | 2865 ± 150 | 1436-786 BCE (95.4%) |
| | | C04* | GX-16333 | Org | 3110 ± 90 | 1604-1583 BCE (0.8%) <br> 1544-1116 BCE (94.7) |
| | | C05* | GX-16330 | Org | 4055 ± 175 | 3089-3056 BCE (0.7%) <br> 3034-2128 BCE (93.7%) <br> 2091-2042 BCE (1.1%) |
| | | C06* | GX-16337 | Org | 5040 ± 100 | 4044-4012 BCE (2.4%) <br> 3999-3640 BCE (93.0) |
| | | C07* | GX-16332 | Org | 6425 ± 130 | 5623-5205 BCE (89.4%) <br> 5173-5070 BCE (6.1%) |
| | | C08* | GX-16331 | Org | 19800 ± 1650 | 26596-18541 BCE (95.4%) |
| | | C09* | GX-16334 | Org | 20700 ± 1600 | 27370-19763 BCE (95.4%) |
| Rivodutri | PDC | C01* | GX-18926 | Org | 2265 ± 80 | 541-95 BCE (94.6%) <br> 73-56 BCE (0.8%) |
| | | C02* | GX-18925 | Org | 3440 ± 90 | 2008-2004 BCE (0.2%) <br> 1971-1517 BCE (95.2%) |
| | | C03* | GX-18924 | Org | 5305 ± 150 | 4446-3794 BCE (95.4%) |
| | | C04* | GX-18923 | Org | 6770 ± 215 | 6077-5307 BCE (95.4%) |
| | | C05* | GX-18922 | Org | 10810 ± 470 | 11787-11764 BCE (0.2%) <br> 11701-11686 BCE (0.1%) <br> 11660-9322 BCE (95.1%) |
| | | C06* | GX-18921 | Org | 13490 ± 235 | 15056-13695 BCE (95.4%) |
| | | C07* | GX-18920 | Org | 15650 ± 680 | 18814-15483 BCE (95.4%) |
| Cava Caporio | Caporio | C01* | GX-17913 | Org | 33300 +6400/-3500 | |
| | | C02* | GX-17916 | Org | >37000 | |
| | | C03* | GX-17914 | Org | >42000 | |
| | | C04* | GX-17915 | Org | >37300 | |






### 4.1 Rivodutri area: Northern Border Fault

The northern Area includes two Sectors of investigation: Campigliano Sector, to the west and Piedicolle / Apoleggia Sector, to the east (Fig. 2a). We excavated 4 exploratory trenches in the Piedicolle / Apoleggia Sector, and one trench in the Campigliano Sector. In the following, we present results from two paleoseismological trenches at the Apoleggia Sector (APO-T1 and APO-T2; Fig. 2b) and one in the Campigliano Sector (CAM; Fig. 2c).



**Figure 2: Study Area along the Northern Border fault: a) simplified geological map; b), c) and d) panels display detailed views on the two studied Sectors with the traces of the capable faults, of the geophysical investigations and the footprints of the excavated paleoseismological trenches; star in Fig. 2a is the Piedicolle trench Site from Michetti et al. (1995)**



The fault scarp we investigated at Apoleggia Sector shows all the characteristics of a typical post-glacial bedrock fault scarp (Roberts and Michetti, 2004, and references therein). The topographic profile of the bedrock fault scarp at the Apoleggia Site shows a minimum post-glacial vertical displacement of 5.2 m (Fig. 3). As discussed in the literature, we assume that this vertical displacement occurred during the last 18 kyr and we derive a consistent post-glacial slip-rate of 0.29 mm/yr at this site. We excavated two trenches at the contact between the Cretaceous-Eocene Scaglia Rossa Fm. and the Early Pleistocene

Villafranchian lacustrine deposits. Trench APO-T1 was 24 m long with a maximum depth of 3 m. We fully covered the excavation walls with detailed photographic survey. However, due to problems in maintaining the stability of the trench walls, we could not prepare a grid for logging. The NE limit of the trench is located a few meters from the bordering fault scarp characterized by the bedrock outcrop. We recognized four stratigraphic units in the APO-T1 trench, reported on the log of Fig. 3a. The stratigraphic units are briefly described as follows: U1a is comprised of white and light gray clays with

calcium carbonate concretions at the base; U1b is a reddish-brownish soil truncated at the top by an erosive surface; U2 is slope debris with angular flint; U3 is a colluvial deposit characterized by loose limestone breccia with angular flint and minor calcareous clasts; U4 is a colluvial deposit characterized by loose limestone breccia with prevalent calcareous clasts and probably very recent brick fragments. APO-T1 shows a man-made hole filled with limestone blocks of various sizes between 15 m and 20 m (each trench is labelled with progressive numbers in meters from the beginning of the excavation;

Fig. 3a). An exploratory well was drilled with a hand auger in the NE end of the trench. The drilling crossed the loose limestone breccia of Unit 2 until it touched the limestone bedrock at a depth of 1.8 m below the bottom of the trench (Fig. 3a), thus constraining the geometry of the fault scarp at depth. Furthermore, APO-T1 shows some clinoforms possibly related to a rollover anticline whose geometry is consistent with that of the bordering fault (Figure 3a). The clinoforms are also visible in the GPR_18 (Figure 3a) located at the same position of APO-T1 (Fig. 2b).

Figure 3d shows the ERT_14 section (see location in Fig. 2b); the main fault, dipping towards the S, separates a sector with resistivity greater than 150 Ωm in the footwall, with a sector of low resistivity in the hanging wall (5-20 Ωm). Toward the S of the ERT_14 profile, another clear change in the resistivity values can be associated with an antithetic fault dipping toward the N (Fig. 3d). Figure 3d shows similar results from the ERT 15, in dipole-dipole configuration, above, and Wenner-Schlumberger, below.

The trench APO-T2 uncovered a sedimentary sequence comprising 5 units that were displaced along normal fault planes. The stratigraphic units are described as follows: U1 is a massive fluvial deposit with yellow (10YR 7/4) silty sands; U2, U3, U4 and U5 are colluvial deposits characterized by a loose calcareous breccia with angular clasts in a silty-clay matrix with different colors from reddish brown, in U2, to dark brown, in U5. Figure 3c shows the interpreted trench stratigraphy photomosaic and the paleoseismic history reconstructed at the Site. The trench crossed a typical earthquake gravity graben

(*sensu* Gilbert, 1897; Slemmons, 1957; Fig. 3e). The main fault plane outcrops a few meters outside the trench, whereas in the trench, a secondary fault shows evidence of repeated movements that occurred during the deposition of Units 2-5. An older earthquake is inferred from the deformation of Units 2 and 3: the top of Unit 2 and some stratigraphic markers within the unit are deformed and inclined by the fault plane. We interpret Unit 3 as a fine-grained colluvial wedge that was



deposited after the first earthquake; this paleoseismic event is chronologically constrained by 14C dates between 11127 and
5512 cal yr BCE, with a minimum offset of 15 cm. A second paleoearthquake is identified from the deformation of Units 3-
5. The top of U5 is undeformed, and the minimum offset of the underlying units is 22 cm; the age of this earthquake is
between 3533 and 2907 cal yr BCE.





**Figure 3: Paleoseismological trench and investigations along the Northern Border fault – Apoleggia Sector (see Figure 2 for the locations of trenches and geophysical line traces): a) APO-T1 trench eastern wall (for details on the cal ages see Table 1) and**



comparison with the GPR 18 and ERT14 lines; b) APO-T2 trench walls; c) GPR20 line; the inset shows a focus on the footprint of the APO-T2 trench (projected); d) ERT15 lines with indicated the footprint (projected) of the APO-T2 trench; (e) model of the gravity graben following the Gilbert's theory of grouped fault scarps in alluvium (redrawn after Gilbert, 1890; Slemmons, 1957).

We excavated CAM trench (see location in Fig. 2c) for a length of 22 m and a maximum depth of 3 m. The stratigraphic units recognized are on the CAM trench log of Figure 4a. The sedimentary sequence exposed in the CAM trench (Fig. 4a) has silts and clays of the Villafranchian series (U1) at its base, presumably related to the Monteleone Sabino Unit (refer to Cavinato 1993 for the basic reference about the local Plio-Quaternary Stratigraphic units). An erosive surface truncates this unit at the top. Loose sands and silts, with lenses of coarse gravel fining upwards to compacted silty clays (U2), lie on top of U1. This unit is interpreted as a colluvial depositional environment. The succession is closed, at the top, by colluvial reddish

silty sands (U3). The beds of the units and their lower and upper limits are crosscut by a series of faults that also induced considerable folds and deformation within the sediments. Three fault strands, belonging to the same deformation zone, have been identified and coded as Fault A, B and C in Figure 4a. The main fault strand is in fact located outside the trench, at an outcrop along a road cut that prohibited direct trenching. At this road cut location, the Villafranchian conglomerates of the Fosso Canalicchio Unit outcrop along a bedrock fault scarp that puts the conglomerates in tectonic contact with recent

colluvial sediments. Logistical limitations did not allow the main fault plane to be directly investigated through a paleoseismological excavation; however, the trench wall exposed a series of subsidiary structures. The main fault plane, in fact, outcrops ca. 30 m west of the trench limit and, projected towards the trench, it runs only a few m to N of the trench. The fault plane dips towards the SE (Fig. 4a) and separates two different units within the Villafranchian lacustrine series (Monteleone Sabino Unit). Two fault strands, synthetic to the main one (i.e., Fault A and C), progressively displaced the

hanging wall block. An antithetic fault (Fault B), slightly oblique to Fault A, delimits a graben that widens toward the east. Overall, the 'Villafranchian' series in the hanging wall of the fault is downthrown and backtilted northward, that is toward the main fault plane, forming an asymmetric half-graben (earthquake gravity graben sensu Gilbert 1897; Slemmons, 1957; Fig. 4c). Faults A and B displaced the top of U1 in all the walls of the trench. The top of U2 along the east wall apparently seals both, while Fault A clearly displaces the same horizon in the west wall, with a cumulative vertical offset of ca. 35 cm. Given

the clay-rich lithology, the limited cumulative vertical offset (i.e., less than 10 cm on a single fault strand) and the gradual transitions between units, we believe that both faults can be considered active during the deposition of the U3 colluvial deposit, and that U2 recorded the ultimate movement younger than 17415-16872 BCE. Fault C, synthetic to the main one but with low-angle bedding, appears to be sealed by U2 along the east wall, while displacing the base of U3 at the west wall. The total vertical offset of Fault C, measured at some levels in the Villafranchian deposits, is ca. 113 cm, while the base of U3

appears to be displaced by only 27 cm. In this case, Fault 3 is also considered to have been reactivated with the last movement during the deposition of U3.

CAM_ERT_02 section in Figure 4b appears to show a slight discontinuity in the resistivity values in its northern sector, along the progressive 90 and 70 m, probably associated with a S dipping fault. The high resistivity values at the base of the





section, along the progressive 70 m, may be associated with the bedrock. Furthermore, a vertical change in the resistivity

values along the progressives 32-45 m may be associated with an antithetic fault dipping towards the N.





a)



b)

c)





**Figure 4: Paleoseismological trench and investigations along the Northern Border fault, Campigliano Sector; CAM trench (see Figure 2 for trench location); (a) photomosaic of the western and eastern walls (for details on the calibrated ages see Table 1); (b) Wenner -Schlumberger ERT 17 line (trace in Figure 2), the location of the CAM trench (projected) is indicated; (c) model of the gravity graben following the Gilbert's theory of grouped fault scarps in alluvium (redrawn after Gilbert, 1890; Slemmons, 1957).**


### 4.2 Cantalice area: Eastern Border Fault

We investigated the Eastern Border fault at two Sectors along the same fault strand (Fig. 5a). We excavated one trench in Sector 1 (Fig. 5b) and three trenches in Sector 2 (Fig. 5c), two of which will be presented in the following section. For the whole dataset, please refer to the Supplementary Material.


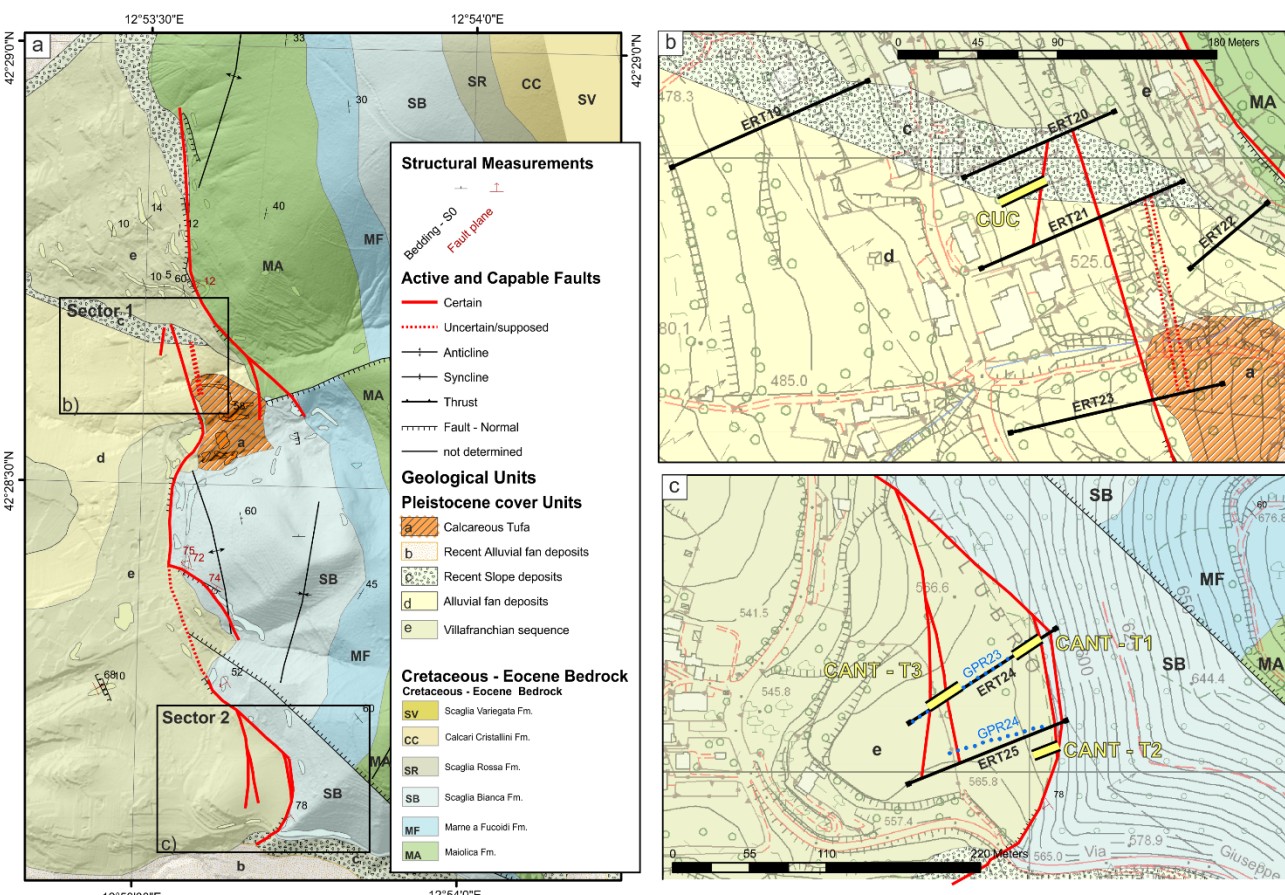

**Figure 5: Study Area along the Eastern Border fault: a) simplified geological map; b) and c) panels display detailed views on smaller sectors with the mapped capable faults, the traces of the geophysical investigations and the footprints of the excavated paleoseismological trenches.**

In Sector 1, we excavated the CUC trench for a length of 14 m and a maximum depth of 3 m. We reconstructed a stratigraphic sequence composed of four units, as described below, from the top to the base of the sequence. The uppermost unit of the stratigraphic sequence (U4) is represented by a matrix-supported conglomerate with well-rounded clasts of

limestone (average diameter 0.5 cm) embedded within a clay-rich matrix (7.5YR 5/4). Remains of roots, bricks and enameled ceramics are dispersed within the matrix.

Unit U3 consists of a matrix-supported gravel made of carbonate clasts embedded within a clay-rich matrix. The matrix of the U3 shows remarkable color variation, from 10YR 5/4; 10YR 4/3 up; 2.5YR 6/6 to base. At the top of the U3, the clasts show frequent Fe (7,5YR 6/8) and Fe-Mn (10YR 3/2) coatings. The amount of Fe-Mn clast coating increases downward. The sub-unit U3a shows local concentration of organic matter and Mn coatings (5YR 2.5/1) are present on clasts.

The unit U2 is made by a highly variable colored clay matrix (10YR 4/4; 10YR 3/1; 10yr 8/2) with rare limestone clasts dispersed within it. The clasts show a maximum size of 3-5 cm, with evidence of partial reworking. In the central part of the trench (i.e. progressive 2-4 meters; Fig. 6a) U2 has been split in several sub-units. From the base: the unit U2d consists of clay-sandy matrix (7.5YR 4/3) with Fe-Mn (7,5YR 5/1) and Fe (7,5YR 5/8) coatings on limestone clasts. Some clasts of the U2d unit are completely decarbonated. Above, the unit U2c is made of sandy-clay (10YR 4/4) with Fe-Mn levels of

concretions (7.5 YR 4/1). Unit U2b is made of a clay matrix (10YR 3/2) with millimeter clasts of limestones and flints. Clasts are covered with Fe-Mn coatings (10YR 3/1). Finally, the unit U2a is made of a matrix-supported gravel of angular 1 cm sized clasts of limestones and flint, dispersed in clay-rich sand (10YR 4/3). Clasts are locally covered with Fe (10YR 8/6) staining.



a)

## CUC trench - South wall

U4 — Matrix-supported gravel (7.5YR 5/4) with limestone clasts up to some cm in size and with common bricks and ceramics dispersed - *historical slope deposits.*

U3 a — Sandy clay with rare limestone and flint clasts up to 1 cm in size; frequence Fe-Mn concretions and clay coatings, more commont downward; downslope the unit is cut by and erosive surface and is laterally passing to a later unit (U3a) made up of light grey sandy clay with rare limestone clasts and common blocks of older soils reworked into the unit.

U2 — Fining upward matrix-supported gravel in a clayey-silty matrix (10YR 4/4) with limestone clasts and common flints; close to the fault the units is organized into four wedge shaped lenses of fining-upward cycles of deposition; FE-Mn concretions are common - *slope deposits and colluvial wedges.*

U1 a — Matrix supported calcareous sandy gravel, light brown, passing upward to a well-developed weathered horizon with frequent zombie stones and flint clasts (7.5YR 5/8); to the top, the unit is truncated by an erosive surface. - *Villafranchian sequence.*

b)

**ERT 20** Wenner - Schlumberger

Resistivity in ohm.m

Model resistivity with topography Iteration 7 Abs. error = 4.8



**Figure 6: Paleoseismological trench and investigations at the Sector 1 of the Eastern Border Fault: a) CUC trench, no vertical exaggeration (southern wall; the footprint in Figure 5b; for details on the cal age determinations see Table 1); b) Wenner - Schlumberger ERT survey (see trace in Figure 5b), the location of the CUC trench (projected) is indicated**

The lowermost unit (U1) is a considerably older sequence, represented by a clast-supported breccia. At its top, unit U1 shows a decimeter-thick distinctive weathering horizon (U1a). The clasts inside the weathered horizon U1a are completely decarbonated. At the top, this horizon is cut by an erosive surface (Villanfranchiano Auct.; Upper Pliocene – Lower Pleistocene). A normal fault, dipping towards W, offsets the sequence exposed at the CUC trench, down-throwing the lowest stratigraphic units of the sequence, i.e., U1-U2. In the hanging wall of the fault, a wedge-shaped colluvial unit crops out (i.e., U2a – d). The fault exposed within the CUC trench shows a polyphasic activity, we observed that a cumulative offset of 87 cm is related to two surface rupturing events. The penultimate event shows an offset of 53 cm, while the last one shows an offset of 34 cm. Both seismic events predate the deposition of the stratigraphic unit U3.

The calibrated dating results obtained from the samples of the CUC trench constrain the last activity of this fault within the VI-V century BCE. In fact, the date of 751-408 BCE refers to the wash facies of the upper colluvial wedge (Unit U2a), which postdates the last surface faulting event. We remark that also the dated sample from Unit 4 gives a similar calibrated age of 401-208 BCE; the whole sequence of slope deposits therefore formed in a few centuries after the observed surface faulting events. It is possible to hypothesize that the fault splay located upslope from the trench that was imaged in the ERT 20 profile (Fig. 6b) also slipped during the same events. The 53 cm offset observed during the penultimate event in the CUC trench, that is relatively large when compared with the other trenches excavated during our investigations, might nevertheless be a minimum value for the coseismic displacement at this site. This would seem to agree with the position of the CUC trench located along the main fault of the Rieti Basin, where we expect the highest displacement during a strong earthquake.

At Sector 2, we investigated a strand of the eastern border fault, where a well-expressed bedrock fault scarp in the Cretaceous Scaglia Bianca Fm. faces a nearly flat paleosurface. The fault scarp we investigated shows all the characters of a typical post-glacial limestone fault scarp (Roberts and Michetti, 2004, and references therein). We acquired two ERT lines (24 and 25; Fig. 5c) that clearly show the tectonic contact between the high resistivity limestone and a package of conductive slope deposits at least 15-20 m thick. To the WSW, the ERT 24 line shows two other abrupt steps in the resistivity units, consistent with a set of antithetic normal faults, bounding the flat paleosurface. Topographic profiles of the bedrock scarp and thickness of Late Holocene to historical slope deposits show a minimum post-glacial vertical offset of 4.5 m (Fig. 7a), hence a slip-rate of 0.25 mm/yr.

The lateral contact between the ERT units, at the ENE tip of the ERT line, well fits with the location of the morphological scarp at the contact between the Cretaceous Scaglia Bianca Fm. and the Villafranchian lacustrine deposits, and with the occurrence of a fault plane, that has been exposed by trench excavation (trench CANT-T1; Fig. 7b). CANT-T1 trench exposes a sequence of Pleistocene Villafranchian units (U1 in Fig. 7b) separated by erosive surfaces, lying directly on a deeply weathered bedrock (U0). The Villafranchian, in turn, is covered by a thick bed of massive colluvial deposits of Late





Glacial to historical age (U5). At the contact between the bedrock and the U1 unit an abrupt step was exposed, marked by the
fold and lowering of U1 toward the hanging wall block. A discrete fault plane has not been observed here, and we were not
able to excavate further in the trench for security reasons. The fault zone is here expressed as a wide and distributed zone of
strain accommodation, due to the highly plastic weathered units exposed. Nonetheless, episodic deformation events are here
confirmed by the geometry of the erosive surfaces, with the erosive surface "es1" being deformed by a first fault movement,
predating the erosive surface "es2", instead. A possible second fault movement, can be inferred based on the abrupt step,
only partially exposed, that the "es2" displays just on the floor of the trench.

Trench CANT-T3 exposes a sequence of Villafranchian units (U1 to U4; Fig. 7c), overlain by a sequence of colluvial beds
with intercalated paleosol horizons. The units are involved in a set of asymmetric folds, which we interpret to be related to
underlying blind normal faulting, based on the features of the ERT-24 line described above. Dated samples in the colluvial
units indicate a Late Glacial to Holocene age. The colluvial beds, steeply dipping toward the mountain front, constitute the
syn-growth sedimentation of the graben-structure with a contemporary infilling of the subsiding block. To the WSW, the
trench also exposes a secondary normal fault in the Villafranchian Sequence, dipping to the SSE, crosscut almost along
strike by our trench, and testifying to complex deformation and hanging wall release faulting in the graben block.

From the features described above, we can constrain several fault movements for this fault strand since Late Pleistocene.
Deformation is distributed into a wide deformation zone with the development of a graben structure and near-surface fault-
related extensional folds.





**Figure 7: Paleoseismological trench and investigations at the Sector 2 of the Eastern Border Fault: a) ERT line with a zoom in on the fault scarp (see trace in Figure 5c); b) and c), paleoseismological trenches T1 and T3 respectively (the trenches footprints in Figure 5c; for details on the cal ages see Table 1).**

### 4.3 Rieti – Santa Rufina area: Southern Border Fault

The third Study Area extends about 5.5 km east of the historical center of Rieti, up to Santa Rufina, a hamlet of Cittaducale (RI), along the eastern part of the Rieti Southern Border Fault. The southern Area includes three Sectors of investigation (Fig. 8). We dug seven exploratory trenches in this Area (Figure 8b-d). Below, we describe the results of paleoseismological





investigations carried out only on the three trenches (TR1, Villa Stoli South and TR3) where we find primary coseismic

elements and, as examples of the structural styles of the fault zone, we show two trenches with only secondary elements

(TR4 and TR5; Figure 8).





**Figure 8: Study Area along the Southern Border Fault: a) simplified geological map; b), c) and d): detailed views on the three studies Sectors (extent is shown in a) with boxes) with the traces of the capable faults and of the geophysical investigations and the footprint of the excavated paleoseismological trenches.**



### 4.3.1 Sector 1: Trenches TR1 and TR5

The 20 m long trench TR1 was excavated (location in Fig. 8b) with a NW-SE orientation where ERT 06 and GPR Line 09 investigations showed clear geophysical discontinuities. Figure 9 shows the result of the inversion of the Dipole-Dipole data acquired along the ERT 06 profile (RMS 9.4%, 4 iterations) using 72 electrodes with an equidistance of 3 m. The
tomographic model shows, in its southernmost part, two significant lateral variations in resistivity. In the same zone, also the GPR 09 profile shows several discontinuities in the electromagnetic characteristics of the shallow part of the terrain (dots lines). TR1 confirmed the occurrence of a fault zone affecting both the Monteleone Sabino Synthem (Early Pleistocene, Upper Villafranchian in Fig. 8), which in this sector is completely pedogenised, and the recent *colluvia* above, outcropping in the upper wall of the excavation. As shown in Figure 9, the trench wall has in fact a ca. 1.5 m thick lower part, separated
from the upper ca 70 cm thick upper part by a step about 1 m wide ("ledge" in Fig. 9).

The overlap between the stratigraphic log and the corresponding section of the GPR 09 profile clearly illustrates how the discontinuities detected by the radar correspond well to those observed in the trench. This result confirms the effectiveness of GPR prospecting in the siting of paleoseismic trenches.

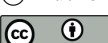



**Figure 9: Southern Border Fault, Sector 1, trench log and geophysical profiles at Trench 1 Site; Above: ERT 6 and Gpr 09 profiles; Below: Stratigraphic log of the W wall of trench 1, accompanied by a legend describing the various pedogenized deposits of the Monteleone Sabino Synthem and the overlying *colluvia* in clear discordance; the photomosaic of the lower west trench wall (progressive 7 m to 17 m) is also shown together with a detail of the major fault localized between progressive 8 and 11m; on the bottom right, the overlap of the stratigraphic log and the GPR profile.**



Figure 9 also presents the stratigraphic log of the W trench wall, accompanied by a legend that describes the various colluvial and soil horizons. Please refer to this legend for details on the stratigraphy visible in the trench walls. Throughout the trench, an ancient and deep pedogenesis characterizes Unit 11. However, in the upper part of the trench Unit 11 is affected by a much more superficial and recent soil horizon (Unit 11a), dated at 7818-7597 BCE. A N60°E trending, 70° NW dipping, fault offsets the Monteleone Sabino Synthem, such that the soil horizons present at its footwall do not correlate

with any of the horizons at its hanging wall; the same fault also displaces the overlying succession of colluvial deposits. We identified this fault as the major fault at the base of the northern slope of the Cappuccini-Villa Potenziani relief.

In the upper wall of the trench, this fault brings in contact the pedogenized Monteleone Sabino Synthem, here characterized by a recent superficial soil horizon (indicated as Unit 11a), with a soil colluvium (Unit 3). The calibrated age of the faulted units provided age ranges of 7818–7597 BCE (Sample C01, Table 1) and 3811 –3701 BCE (Sample C02), respectively. This

means that a surface faulting event occurred after 3811-3701 BCE (event TR1-2 in Fig. 15).

In the lower wall of the trench, the fault brings Unit 11 in contact with a colluvium (Unit 6) which, due to its geometry, we interpret as a colluvial wedge. This colluvial wedge returned a calibrated age of 20919–20458 BCE (Sample C05) and, considering its thickness, it is possible to estimate a surface dislocation of at least 15 cm necessary for it to be deposited. This event occurred just before the deposition of the colluvial wedge, then before 20919–20458 BCE (event TR1-1 in Fig.

410 15).

In sector 1 (Fig. 8b) we also excavated the trench TR5, providing evidence of several tectonic dislocations of the Monteleone Sabino Synthem deposits (Fig. 10). We interpreted these structures as secondary faults, presumably linked to a main fault located just to the north. However, it was impossible to verify this hypothesis; we were not able to extend the exploratory trench further N, due to the presence of an aqueduct.



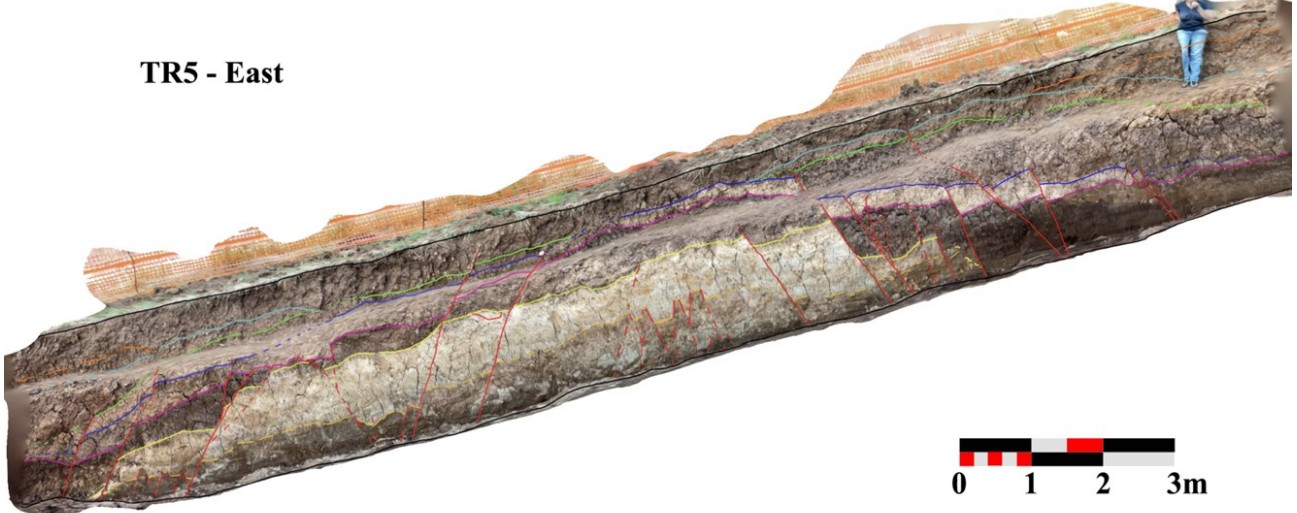


**Figure 10: Southern Border Fault, Sector 1, trench 5 showed the Monteleone Sabino deposits dislocated by a dense grid of faults; the figure shows a portion of the east wall of the 77 m long trench, no vertical exaggeration.**

### 4.3.2 Sector 2: Villa Stoli South Trench

The second Sector under investigation is the Villa Stoli area, where ERT 02 and GPR 28 have been carried out (Fig. 8c; see
Supplementary Data for GPR 28). In this case, only the tomography detected discontinuities due to tectonics. They are, in the southern part of the profile, corresponding to a wooded area where the GPR was not applied (Fig. 11). The anomalies detected, indeed, in the central-northern part of the ERT profile (Fig. 11) and in the GPR profile, correspond to the travertine slab found in the Villa Stoli Nord trench (see geology in Fig. 8c).

We excavated the Villa Stoli South trench for a length of 40 m, a depth of about 2 m and a width of 1 m. In the uphill, S part
of the trench, we uncovered three major discontinuities, which were also detected by the ERT 02 survey. Figure 11 shows the stratigraphic log with tectonic elements.

The first discontinuity of the fault zone FZ1 offsets the Lower Pleistocene cemented breccia (Fosso Canalicchio Synthem, FC) and, with decimetric throw, also all the overlying deposits, until it is sutured by the current organic soil. Eight meters downstream, the elements of the FZ2 places the FC Synthem breccia (Unit 2) in contact with metric blocks of conglomerates
associated with sandy-clayey sediments, interpreted as belonging to the Monteleone Sabino Synthem (MS). FZ2 fault zone upward cuts the overlying colluvium made of decarbonated dark reddish-brown soil (2.5YR 3/4) (Unit 19 in Fig. 11), whose AMS calibrated age (sample C12) is 21261-20961 BCE (95.4%). The faulted Unit indicates the occurrence of an event post 21261-20961 BCE. This event is possibly correlated with the one observed in Sector 1, TR1, where a colluvial wedge dated back to 20919–20458 BCE. The two events TR1-1 and VS-1 in Figure 15 might therefore indicate the same





paleoearthquake. Starting from this point, the soil colluvium (Unit 19) thickens downwards, resting on a decarbonated clayey horizon (Unit 6 in Fig. 11), developed on the clay-rich facies of the MS Synthem.

We identified another fault zone (FZ3) ten meters further downhill, toward the north. This fault offsets the interface between the soil colluvium (Unit 19) and the pedogenized clay-rich horizon (Unit 6) with a throw up to 30 cm. The 14C calibrated age of the younger deposits dislocated is 11835-11630 BCE (85.0%; Sample C16). This presumably means that a second

seismic event occurred after this date.







**Figure 11: Southern Border Fault, investigations at Sector 2 – Villa Stoli; Above: ERT 02 profile; Middle: Photographic mosaic and stratigraphic log of the southern and upper part (first 10 m) of the E wall of the Villa Stoli Sud trench, including fault zones**
**Fz1 and Fz2, and stratigraphic log superimposed on the southern part of the ERT 02 profile; Below: Stratigraphic log of the whole Villa Stoli South eastern trench wall.**

### 4.3.3 Sector 3: Trenches TR3 and TR4

We selected the location of the trench TR3 based on geological and geomorphological evidence, without carrying out geophysical investigations. It was dug at the main slope break on the northern slope of the Castellaccio relief (Fig. 8d).

The excavation began where a fault outcrops in the cemented breccia of the Fosso Canalicchio Synthem (basal Lower Pleistocene in age), where the trench walls revealed open fractures filled by colluvial deposits. The breccias present sub-angular to angular clasts with generally centimetric up to decimetric diameters. The fracture, visible at progressive 1 m of the E trench wall (Fig. 12), is filled by brown paleosols that were sampled at different depths. The calibrated ages of the lower sample C06 and of the upper sample C07 resulted in 4988-4797 BCE (95.4%) and in 326-424 CE (78.6%) respectively.

Since the filling of a seismic-induced open fracture supposedly occurs just after the event, an event should be dated just before 326-424 CE (event TR3-3 in Fig. 15) and another just before 4988-4797 BCE (event TR3-1 in Fig. 15).

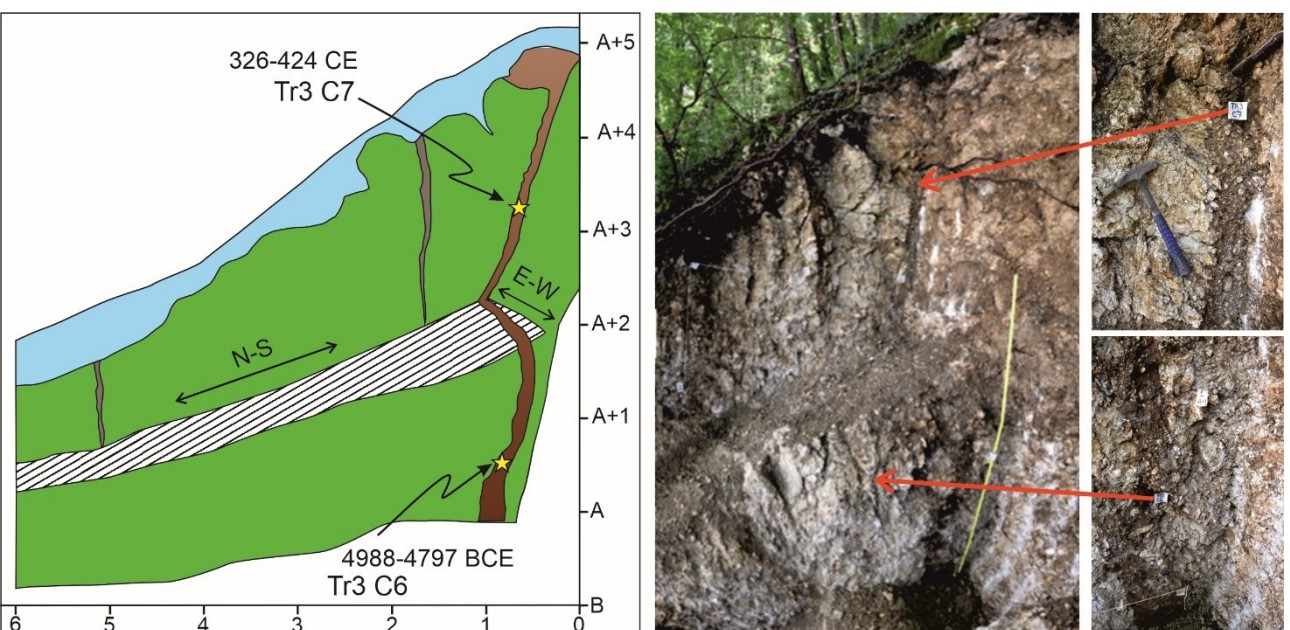

**Figure 12: Southern Border Fault, Sector 3; Left: tectonic fracture outcropping at the S end of the trench TR3, filled by colluvial deposits sampled in TR3 C7 (C07) and TR3 C6 (C06); calibrated C14 ages are, respectively, 326-424 CE (78.6%) and 4988-4797**
**BCE (95.4%); Right, log of the first 6 m of the E trench wall; "ledge" is a bench due to the backhoe excavation**





From progressive 11 m toward N, there is an over one meter wide fault zone which brings in contact the Fosso Canalicchio Synthem with the overlying Monteleone Sabino Synthem (Fig. 13), the fault zone can be distinguished into two fault subzones.

The fault zones involve recent paleosols and colluvial deposits. Among them, Units 6 and 7 (Fig. 13), located only at the hanging wall of the FZ1, are evidently backtilted toward the fault. Unit 7 is a brown paleosol (7.5YR 5/4) developed on the sands of the Monteleone Sabino Synthem, with a calibrated age 4171-4036 BCE (60.9%; Sample C01). Above Unit 7 there is a soil colluvium (Unit 6) that contains small calcareous clasts, with a calibrated age 7729-7589 BCE (95.4%; Sample C02). Going stratigraphically up section, Unit 5 is a dark brown colluvium (10YR 3/3) with a calibrated age 3711-3627 BCE (87.7%; Sample C03), and Unit 4 is made of sub-angular-angular deposits, multi-centimetric in size, with a pedogenized matrix, with a calibrated age 3243-3102 BCE (57.2%; Sample C04). The most downstream part of Unit 4 is thickened at the hanging wall of the FZ1 (yellow star in the picture detail of Fig. 13) and because of this, it is interpreted as a colluvial wedge that formed following the surface faulting event that tilted Unit 6 and 7. In this hypothesis, the underlying dark brown colluvium (Unit 5), interrupted by FZ1, must have been deposited before the seismic event. The coseismic fault slip is 13 cm. All the above considerations suggest the occurrence of a seismic event between 3627 and 3102 BCE. The age constraints for this event are very similar to those available for the last event detected at APO-3 trench near Apoleggia along the N Border Fault, on the opposite side of the Rieti Basin.

Moving N, the Monteleone Sabino Synthem outcrops with subvertical layers, parallel to the fault, probably because they were dragged along the fault itself. These deposits are characterized by alternations of sandy and silty layers, rarely interspersed with layers of fine gravel (millimetric clasts). Also, a system of mesoscale tectonic lineaments was recognized along the wall of the trench, which are generally subparallel to the bedding. Therefore, we assume bed-parallel tectonic slip occurred here.

Towards progressive 24-26 m, the inclination of the Monteleone Sabino Synthem deposits decreases and the contact with a pedogenized colluvium with clasts altered by pedogenesis (ghosts of clasts or "biscuit" clasts) has a slope of approximately 30° towards the valley (see Fig. 13).

Above the pedogenized colluvium, there is sharp contact with colluvial deposits characterized by frequent sub-rounded calcareous clasts, with decimetric diameter, immersed in an abundant soil matrix. This contact, which is dipping about twenty degrees towards the valley, becomes abruptly sub-horizontal in the N terminal zone of the trench.




**Figure 13: Southern Border Fault, Sector 3; portion (from progressive 8 m northwards to 26 m) of the E wall of the TR3 trench:**
**detail and log of the fault zone putting in contact the Fosso Canalicchio Synthem (Unit 10) with the Monteleone Sabino Synthem (Unit 9); the stratigraphic log shows the deposits, the sampling points for AMS dating and the fault zones Fz1 and Fz2.**

In trench TR4, located north-east with respect to trench TR3 (see Fig. 8d) and 26 m long, we found the deposits of the Monteleone Sabino Synthem dipping toward north-west, with slopes between 40° and 70°. Between the progressive 18 m and 19 m, gray silts diminish from the top to the bottom of the trench wall and are delimited by two fault surfaces (Fig. 14).
In this case, as in TR3, bedding-parallel slip is assumed. We observe that the faults are directly sealed by ploughed soil, and no deposits younger than lower Pleistocene are involved in the deformation.



**Figure 14: Southern Border Fault, Sector 3; trench TR4, the bedding of the Monteleone Sabino Synthem dips between 40° and 70° toward NW along the 26 m long trench; between the progressive 18 m and 19 m, gray silts diminishing from the top to the bottom of the trench wall are delimited by two faults.**

## 5 Discussion

Before entering into the discussion of the data, it is necessary to highlight some "disclaimers". In fact, the interpretation and correlation of paleoseismic data obtained during the study campaign on the Rieti Basin capable faults conducted in 2021 - 2022 is necessarily limited due to the peculiar logistic and the stratigraphic/geomorphic field setting.

First, we investigated selected areas identified based on administrative criteria. We conducted geophysical prospecting and trenching on "active and capable faults" as defined by the Microzoning Guidelines of the Italian Government (Commissione Tecnica per la Microzonazione Sismica, 2015). We worked in the framework of the "Scientific Collaboration Agreement between the Extraordinary Commissioner for Reconstruction and the National Institute of Geophysics and Volcanology (INGV)" for the redefinition of the Attention Zones of Active and Capable Faults emerging from the seismic microzonation



studies carried out in the Municipalities affected by the seismic events that occurred starting from 24 August 2016 (https://sisma2016data.it/faglie-attive-e-capaci/). For this reason, we aimed our investigations to specific technical targets, and associated progressive deadlines, for the timely definition of the avoidance belts where the construction of new buildings and the reconstruction of buildings damaged by the 2016-2017 seismic sequence had to be prevented.

Second, in the Rieti alluvial and lacustrine plain, literature data clearly show that the Late Holocene sedimentation rates are
tenfold the expected fault- slip rates (Guerrieri et al., 2006; Archer et al., 2019; Brunamonte et al. 2022). We confirmed this conclusion also through our exploratory trenches in the N Study Area of the Rieti Basin. In the Rivodutri Area, we located two trenches (Villaggio Santa Maria and Piedicolle trenches) in the lower part of the slopes, where ERT profiles suggested the presence of faulting at shallow depth. In both trenches, we obtained AMS ages younger than 2000 years BP from colluvial and alluvial deposits at 3 m below the ground surface. The presence of Roman pottery at the base of the trench
walls confirmed AMS dating. Therefore, our trenches did not reach the possible faults beneath the ground surface because of the excessive thickness of very young historical deposits. In general, in the Rieti plain it is not possible to find suitable sites for paleoseismic trenching; and it is not possible to expose faulted Holocene stratigraphy in the high-resolution lacustrine sediments. We excavated all our 13 trenches at or near the bedrock-slope deposits contact, typically delimiting the reconstruction of paleoseismic events with less well constrained chronologies and estimates of coseismic displacement.
Having this in mind, our new dataset, also compared with past paleoseismic analyses from Michetti et al. (1995), provides relevant information on the paleoearthquake dates and magnitude, as discussed in the following sections.

**5.1 Post- Last Glacial Maximum (post-LGM) fault slip-rates and limestone bedrock scarp evolution in the Rieti Basin landscape**

Data from our trench investigations confirmed the general model of post-LGM and Late Glacial to Holocene evolution for
bedrock fault scarps in the Meso-Cenozoic pelagic marly limestones of the Umbria-Marche-Sabina Sequence. As extensively discussed in the literature (Bosi, 1975; Serva et al., 1986; Blumetti et al., 1993; Lorenzoni et al., 1993; Michetti et al., 1995; Roberts et al., 2025), the morphological evidence of Holocene tectonic and seismic activity is very clear along the mountain fronts carved in the Meso-Cenozoic Abruzzi carbonate platforms. The surface processes on the coherent pure limestone or dolostone bedrock from Mesozoic carbonate platform environments, which underwent slope recession during
the Last Glacial, are now very limited and cannot cancel the morphological effects of tectonic movements. The same is not true for the Rieti Basin and other Quaternary basins in the Umbria-Marche-Sabina sector (such as the Norcia, Leonessa and Colfiorito basins; Messina et al., 2002; Mildon et al., 2022), where the presence of forest cover on the slopes indicates the development of pedogenetic and colluvial processes on the material supplied by the Mesozoic marly-clay-rich units.

Moreover, human impact on the mountain slopes through deforestation and agriculture generated a significant increase in the
sedimentation rates of slope deposits since the early stage of the Roman Empire, and thereafter (Lorenzoni et al., 1993; Michetti et al., 1995; Borrelli et al., 2014). Stratigraphic and geoarchaeological data at La Casetta site clearly document this



anthropogenic impact, with meters of Roman to Middle Ages slope deposits containing abundant and well dated pottery remains.

This is therefore the reason why bedrock fault scarps in the Rieti Basin are discontinuous and relatively subdued. However,
we successfully investigated two of these bedrock fault scarps, one along the N border fault at Apoleggia sector, and the second along the E master fault at Cantalice sector. Trenching and geophysical prospecting in the Cantalice area, Sector 2, shows faulted historical colluvial deposits against the Scaglia Fm. fault plane. The measured slip-rate of 0.25 mm/yr is a minimum, because our trenching did not reach the base of historical slope sediments. We draw a similar conclusion also at Apoleggia. Here we observed a synthetic fault splay in the hanging wall of the bedrock fault scarp. Therefore, the measured
slip-rate of 0.29 mm/year is also a minimum estimate.

Nevertheless, these values are meaningful because we have both morphological and stratigraphic constraints. These new post-glacial fault slip-rate data confirm the previous estimates available in the Rieti Basin. Roberts and Michetti (2004) measured 0.27 mm/yr at a site N of Rivodutri along the Rieti Eastern Border Fault; and Michetti et al. (1995) documented 0.4 mm/yr over the Holocene at the Piedicolle trench site along the Northern Border Fault.

Several Late Quaternary faults in Central Italy share similar post-glacial slip-rates (Roberts et al., 2025; Lombardi et al., 2025). For instance, the Mt. Morrone Fault (Puliti et al., 2024); the Paganica Fault that generated the April 6, 2009, Mw 6.0 earthquake (Cinti et al., 2011) the Montereale Fault which ruptured during the January 16, 1703, Mw 6.0, earthquake (Cinti et al., 2018) and the Mt. Vettore–Mt. Bove normal fault system source of the Mw 6.5 Norcia earthquake of October 30, 2016 (Cinti et al., 2019).

**5.2 The paleoseismic history of the Rieti Basin: a possible pattern in the earthquake sequence?**

We compiled the paleo-earthquakes identified in the trenches with those ones recognized by Michetti et al. (1995) (i.e., the Piedicolle and La Casetta trenches, PDC and CAS, respectively, with original Geochron Lab. Radiocarbon dates recalibrated using the IntCal20 curve) and summarized all the data in the space-time diagram in Figure 15.

Red bars are the time spans of the calibrated ages, defining the terminus *ante-* and/or *post-quem* for earthquakes (see Table
2). Some paleo-events are tightly constrained in time (e.g., CUC-2), whereas in other cases we have less robust chronological constraints, and in some cases can only bracket the timing of the paleoseismic events.

We highlighted five events, here coded as A to E, which are chronologically well-constrained (i.e., black boxes in Figure 15), plus one less-constrained additional event (F in Figure 15). The six events do not overlap in time, thus possibly representing distinct earthquakes. Correlation of paleoseismic events recognized in different trench sites are inherently
speculative in nature. Moreover, for the Rieti Basin we consider these correlations unlikely, given the large chronological uncertainty. Nevertheless, we think that these correlations should be taken seriously into account, as they depict a conservative scenario in terms of reconstructing the maximum credible earthquake magnitude and associated seismic hazard. Below we summarize the evidence driving our interpretation, supplemented with constraints from historical catalogues, and we discuss potential ruptures of the whole Rieti Basin Fault segment.





Event A is constrained at trench TR3 in the SE margin of the basin; the coseismic movements that occurred over time along the identified fault (Fig. 12) cyclically determined the reopening of the fracture, which was repeatedly filled with *colluvia*. The younger age obtained from the *colluvia* trapped in the fracture is 326-424 CE. No event in the historical seismic catalogue matches with our Event A.

A few centuries before, a strong event is reported to have occurred in the Rieti area in 76 BCE (Guidoboni et al., 2018, 2019). Obsequens Iulius (1910) stated:

> *"Reate terrae motu aedes sacrae in oppido agrisque commotae. Saxa quibus forum stratum erat, discussa. Pontes interrupti. Ripae [prae] labentis fluminis in aquam provolutae, fremitus inferni exauditi et post paucos dies, quae concussa erant, corruerunt"*
>
> (Due to the earthquake, the sacred buildings in the town and the fields were moved. The stones with which the market was laid were broken up. Bridges broken. The banks of the flowing river were thrown into the water, the roar of hell was heard, and after a few days those that had been shaken collapsed).

**Table 2: Attributes summary of the paleo-earthquakes identified in the trenches excavated in the Rieti Basin; the "notes" column refers to the possible ruptures of the whole Rieti Fault segment as described in the text; asterisks mark the events that "control" Event A, B, C, D, E; F; the others are the possible correlations.**

| Trench | Earthquake | Younger than | Older than | Offset | Notes |
|---|---|---|---|---|---|
| TR3 | 3 | | 326-424 CE | | A* |
| TR3 | 2 | 3711-3627 BCE | 3243-3102 BCE | 13 cm (min) | C |
| TR3 | 1 | | 4988-4797 BCE | | D* |
| TR1 | 2 | 3811 –3701 BCE | | 30 cm | A, B, C |
| TR1 | 1 | | 20919–20458 BCE | 15 cm (min) | F* |
| VS | 2 | 11835-11630 BCE | | | B, C, D, E |
| VS | 1 | 21261-20961 BCE | | | B, C, D, E, F |
| CUC | 2 | 591-408 BCE | 401-351 BCE | 53 cm | B* |
| CUC | 1 | | 591-408 BCE | 34 cm | C, D, E, F |
| CAS | 1 | 5623-5205 BCE | 3999-3640 BCE | | D |
| PDC | 2 | | 1971-1517 BCE | 100 cm | C, D, E, F |
| PDC | 1 | 4446-3794 BCE | | 90 cm | A, B, C |
| APO T2 | 2 | 3533-3370 BCE | 3110-2907 BCE | 22 cm (min) | C* |





| APO T2 | 1 | 11127-10795 BCE | 5664-5512 BCE | | E* |
| CAM | 1 | 17415-16872 BCE | | 27 cm (last event) | A, B, C, D, E |

In the Italian seismic catalogue, the 76 BCE event has an Epicentral Intensity X and an intensity-derived magnitude Mw 6.4. (Guidoboni et al., 2018). Guidoboni et al. (2018, 2019), based on the only historical source available, place the epicenter in the town of Rieti, but its exact location could be anywhere in the basin area, or in the SE part of the basin, like the M 5.5

event occurred in 1898 (Rovida et al., 2022; Comerci et al., 2003). Chronological constraints obtained from the trenches indicate that the earthquakes identified in TR1 (S margin) or PDC and CAM (N margin) overlap with the 76 BCE, even though the dating uncertainties are high.

Regarding the extension of faulting associated with Event A, and possible involvement of multiple fault splays in the Rieti Basin, we note that paleo-earthquakes consistent with the chronology of Event A are found in trenches TR1 (S margin of the

basin), PDC and CAM (N margin). We are indeed quite confident that Event A did not produce faulting at APO T2 Site, since the top of Unit 5 seals the deformation there and younger deposits are not faulted (Fig. 3).

Event B is tightly constrained at trench CUC (E margin of the basin); the earthquake occurred between 591-408 BCE and 401-351 BCE, with an offset of 53 cm. A tentative hypothesis about a rupture affecting the E master fault and multiple fault splays in its hanging wall is supported by paleo-earthquakes identified at trenches TR1, VS, PDC and CAM; however, like

Event A, such correlations seem highly speculative.

In the time span between 3 and 5 kyr BCE, all of the three investigated margins of the basin moved: the N margin hosted Event C, constrained at APO T2, dating between 3533-3370 BCE and 3110-2907 BCE and with a minimum offset of 22 cm. Trenching at La Casetta site shows evidence for the movement of the E master fault, dated between 5623-5205 BCE and 3999-3640 BCE. The movement of the S margin is constrained at TR3 Site, where two earthquakes have been recognized:

the one dated between 3711-3627 BCE and 3243-3102 BCE has a good overlap with Event C, while the earthquake dated at 4988-4797 BCE is tightly constrained and identified as Event D. We can confidently claim that Events C and D are two distinct earthquakes, since the date ranges do not overlap. Therefore, Events C and D likely did rupture the whole length of the ca. 21 km long Rieti Basin with a Mw in the order of 6.5. This is consistent with coseismic displacement of more than 50 cm observed at CUC (Event D, CUC1; Fig. 15), and in the order of 1 m observed at PDC (Events C and D, PDC1 and

PDC2; Fig. 15).

Event E activated the N margin and is broadly constrained at APO T2 Site (date range between 11127-10795 and 5664-5512 BCE). Like for the other events, in this case weak correlations with movements on the other margins of the basin are possible.



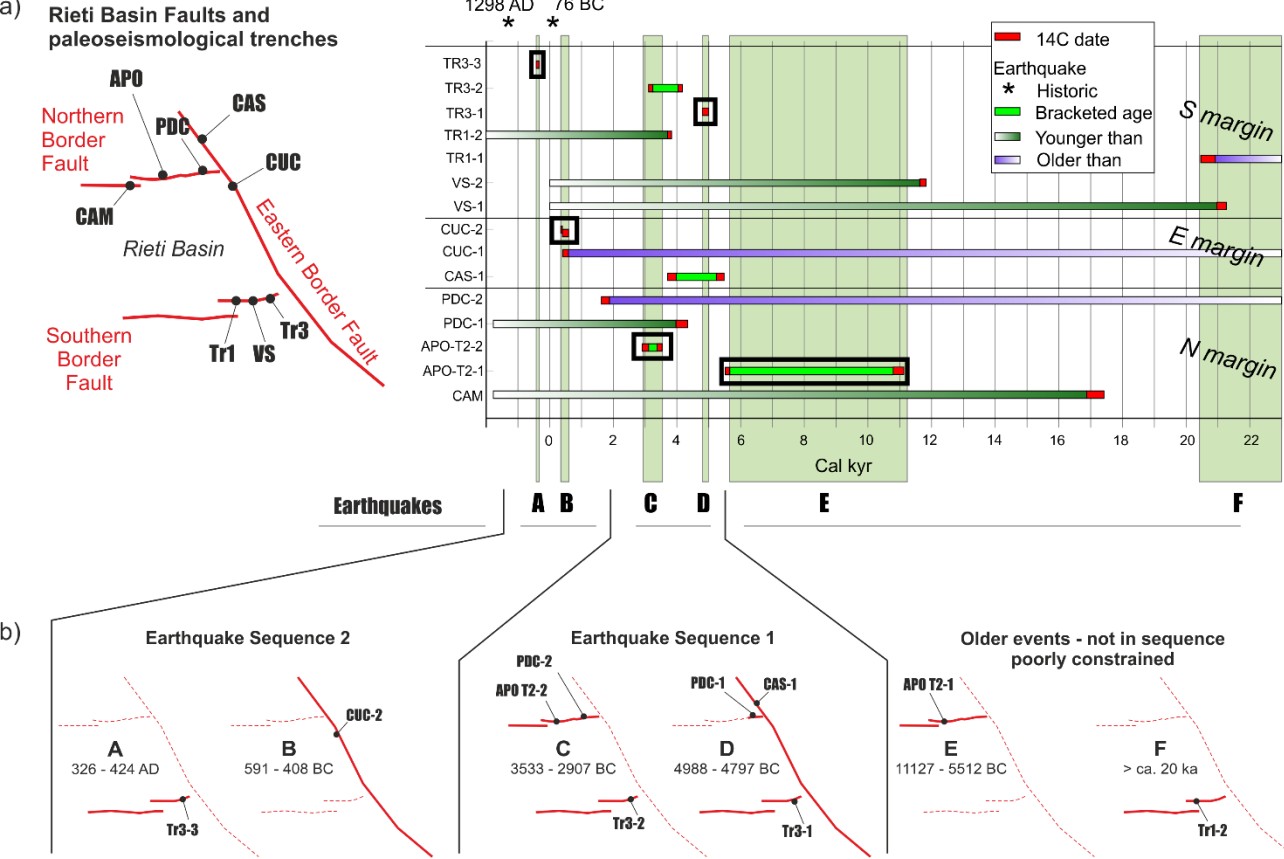

**Figure 15: a) Space-time diagram at the basin scale; on the ordinates the acronym of the earthquakes recognized in each trench (from Table 2); each horizontal bar represents the temporal constraints to the occurrence of each earthquake; black boxes indicate well-constrained events, and the green transparent bars are highlighting possible correlations across trenches; b) summary of the sequence of fault ruptures along the basin-bounding faults, based on well-constrained events (thick continuous lines indicate fault ruptures; thin dashed lines, not assessed).**

Finally, Event F involved the S margin of the basin; it is constrained at TR1, with an estimated age older than 20919–20458 BCE. This event is also responsible for the dislocation of the colluvial Unit 19 dated 21261-20961 BCE in the trench of Villa Stoli (Sample Ri12; event VS1).

Figure 15b summarizes the overall picture emerging from this complex pattern of fault ruptures, given the uncertainties of both the age dating and possible across-trenches correlations. Seven earthquakes have an age bracketed in time, three referring to the S margin (trench TR3), two on the E margin (events CUC-2 and CAS-1) and two on the N margin (trench APO T2). The average recurrence interval is of a few millennia on all the segments; given fault length and the long-term slip rate, the recurrence intervals obtained in our study are consistent with observations on a global scale (Mouslopoulou et al., 2025).

If we only rely on the chronologically well-constrained events correlated across-trenches, these indicate two clusters of events, which we label as Earthquake Sequences 1 and 2 (Fig. 15b).



Earthquake Sequence 1 started with the rupture of the main fault bordering the basin, the Eastern Border Fault, and of the Southern Border Fault around 4988 – 4797 BCE (Event D). Possibly, also the easternmost tip of the Northern Border Fault ruptured, as recorded at the PDC trench (Michetti et al., 1995). The sequence ended up with the rupture of the Northern and Southern Border Faults around 3533 – 2907 BCE.

After an apparent period of quiescence, the inception of Earthquake Sequence 2 is dated 591 – 408 BCE, when the Eastern Border Fault ruptured with a major earthquake (Event C) that was later followed by the movement of the Southern Border Fault at 326 – 424 CE.

The general recurring scheme here is that for each sequence, a main rupture of the Eastern Border Fault is later followed by movements on the other two structures.

The structural setting of the Rieti Basin might shed some light on such behavior. In fact, the basin is bounded to the east by a main west-dipping fault, whereas relatively short orthogonal faults bound it to the north and to the south. The latter faults are constrained within the hanging wall block of the main Eastern Border Fault. Such a fault architecture can be interpreted in terms of a hanging wall release faults model (*sensu* Destro, 1995; Figure 16a). The role of release faults is to accommodate the hanging wall deformation resulting from a bow-shaped profile of cumulative slip along a fault strike. If a significant

displacement gradient exists along the fault strike, then resultant orthogonal extension can be accommodated by release faults. This is the case of the extensional faults in the Central and Southern Apennines, where structural inheritance from previous tectonic phases (Capotorti and Muraro, 2024) results in relatively short faults with high displacement gradients (e.g., Roberts and Michetti, 2004; Papanikolaou and Roberts, 2007; Porreca et al., 2020).

In this line, the proposed rupture sequence would descent from the entire rupture of the Eastern Border Fault: the resulting

stress loading on the other two receiving faults would result in a later re-adjustment of the strain field by means of one or more earthquakes on the hanging wall release faults (e.g., Mildon et al., 2017; Valentini et al., 2024).

The architecture of the faults bordering the Rieti basin is apparently odd, with faults systems striking at ca. 60º-120º each other and are apparently hardly compatible with a single extensional phase, as previously discussed in literature. Nonetheless, other well-documented examples worldwide (Figure 16b-e) highlight that normal faulting events can result in a

surprisingly complex pattern of deformation and surface ruptures, with faults trending at high angles to each other. This mainly results from i) oblique extension and/or ii) the presence of inherited structures partly misoriented with respect to the orientation of the stress field, but may still be compliant for future re-activations.

As a general consideration, the rupture of the whole eastern master fault of the Rieti Basin seems to be the only scenario compatible with the Mw 6.4 estimated by the CFTI5Med catalogue (Guidoboni et al., 2018, 2019) for the 76 BCE event, or

with the Mw 6.26 estimated by the CPTI15 catalogue (Rovida et al., 2022) for the 1298 event. In summary, the paleoseismic history described above suggests a maximum magnitude in the order of Mw 6.5. We discuss below that this interpretation agrees with the available data from surface faulting events that occurred in Central and Southern Italy during the instrumental era, and in particular over the past 45 years.



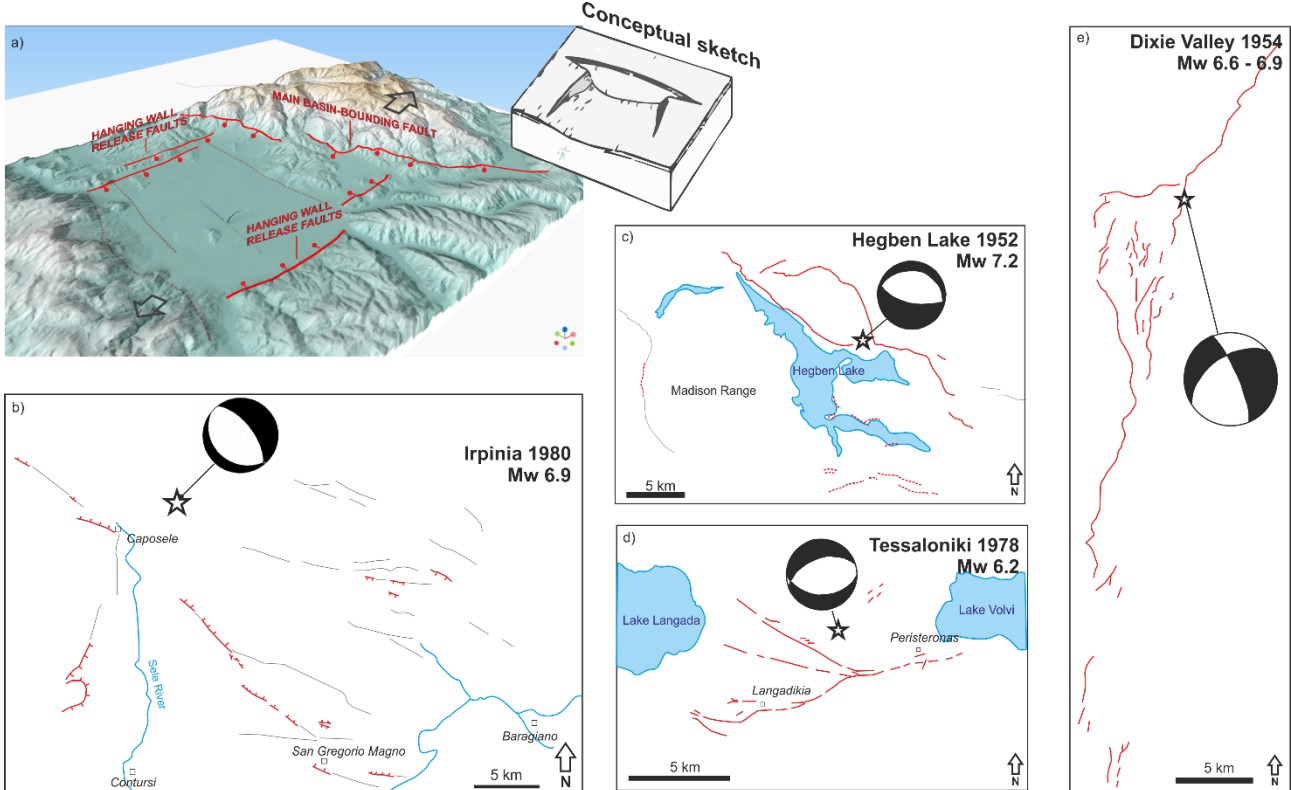

**Figure 16: a) a 3D perspective on the Rieti basin with the studied faults and a conceptual scheme (after Destro, 1995, modified) illustrating the structural interpretation for the area, with the E Fault as Master Fault of the Rieti Basin and the N and S Faults as hanging wall release faults; b) to e): the complexity of the surface fracture pattern occurred during well-documented ruptures accompanying Mw 6.2 – 7.2 normal faulting earthquakes in Italy, Greece and the US Basin and Range Province, with the calculated moment tensor solutions (Doser, 1984; 1986; Liotler, 1989; Giardini, 1993).**

## 5.3 Fault displacement hazard in the Rieti basin

The paleo-earthquakes identified in the Rieti basin show offsets in the order of few cm to 1 m (Table 2). We consider these values consistent with the seismic potential of the region. In a broader perspective, the Rieti basin belongs to the seismotectonic setting of the Central-Southern Apennines, which in the last decades were repeatedly hit by groundbreaking earthquakes.

As illustrated in Table 3, the Colfiorito 1997, L'Aquila 2009, Amatrice 2016 and Visso 2016 earthquakes had magnitude in the Mw 6.0 range and resulted in maximum displacement of 20 cm (Vittori et al., 2011; Boncio et al., 2012; Pucci et al., 2017). The much stronger Fucino 1915, Irpinia 1980 and Norcia 2016 earthquakes (Mw 6.5-6.9) instead generated metric displacements (max values 210 cm; Pantosti and Valensise, 1990; Michetti et al., 1996; Villani et al., 2018).





**Table 3: Ground rupture parameters for recent earthquake surface faulting along normal faults in the Central and Southern Apennines. SRL = Earthquake Surface Rupture Length**

| Earthquake (dd/mm/yyyy) | Mw | SRL (Km) | MAX Disp (m) | Source |
|---|---|---|---|---|
| Fucino 13/01/1915 | 7.0 | 27 (36)* | 1.0* | Michetti et al., 1996; Galadini and Galli, 1999* |
| Irpinia 23/11/1980 | 6.9 | 30 | 1.3 | Pantosti and Valensise, 1990 |
| Colfiorito 26/09/1997 | 6.0 | 12 | 0.08 | Guerrieri et al., 2010 |
| Sellano 14/10/1997 | 5.6 | 1.7 | 0.04 | Guerrieri et al., 2010 |
| Lauria 09/09/1998 | 5.6 | 0.2 | 0.02 | Michetti et al., 2000 |
| L'Aquila 06/04/2009 | 6.1 | 13.0 | 0.15 | Vittori et al., 2011 |
| Amatrice 24/08/2016 | 6.0 | 5.2 | 0.20 | Pucci et al., 2017 |
| Visso 28/10/2016 | 6.0 | 7.0 | 0.4 | Scognamiglio et al., 2018 |
| Norcia 30/10/2016 | 6.5 | 21.9 | 2.4 | Villani et al., 2018 |

As already pointed out, the hypothetical rupture of the whole Rieti Fault would result in a surface faulting length of ca. 21
690  km. This is consistent with the coseismic ruptures in Table 3, in particular with those observed during the October 30, 2016, Mw 6.5, Norcia earthquake.

As a further speculation, we observe that in case of Mw 6.5 events, surface ruptures along the W margin of the Basin would not be seen as a surprise. In fact, the 2016 Mw 6.5 Norcia earthquake generated surface faulting several km west of the Mt. Vettore master fault, including antithetic ruptures that affected the San Benedetto subsurface road tunnel located ca. 8 km
west of the master fault (Galli et al., 2020).

On the other hand, Table 3 shows that the threshold for surface faulting during shallow crustal normal faulting earthquakes in the Central - Southern Apennines is ca. Mw 5.6. Again, this is consistent with observations from recent moderate magnitude events in the Rieti Basin. The June 27[th], 1898, Santa Rufina event (Mw 5.5; Io VIII-IX MCS; Comerci et al., 2003) generated ground fractures E of Rieti. According to Moderni (1899), near Santa Rufina "seven large and long cracks"
had formed in 1898, parallel and close to each other; while five similar long cracks had formed 2-3 km W of Cupaello. However, during the December 31[st], 1948, Rivodutri event (Mw 5.3; Io VIII MCS; Bernardini et al., 2013), near the NE border of the basin, we have no report of ground fractures.



## 6 Conclusions

In this paper we present new insights regarding the paleoseismic history of the Rieti Basin in Central Italy, derived from the
excavation of 17 paleoseismic trenches. Our work represents the first comprehensive characterization of the seismic hazard
of the Rieti Basin, which so far has been underdocumented compared to other areas of the Italian Apennines.

Through extensive fieldwork and analyses, we identified and chronologically constrained up to 15 paleo-earthquakes that
generated surface faulting on three sides of the box-shaped Rieti basin during the last ca. 20 ka. These results provide a
baseline of tectonic activity in the region that was not known prior to this work.

Considering the spatio-temporal distribution of the faulting events, we propose a temporal development characterized by the
earthquake rupture of the Eastern Border Fault, followed by seismic events either on the northern or southern border faults.
This pattern is consistent with the structural architecture of the basin, which comprises two sets of nearly orthogonal faults.

Our results indicate that the maximum credible earthquake in the Rieti Basin is in the order of magnitude Mw 6.5, which is
consistent with the general setting of the Central Apennines. Given the resolution of chronological constraints obtainable
with radiocarbon dating techniques in paleoseismic trenches, we cannot disentangle the occurrence of a single earthquake as
compared to multiple earthquakes occurring over a short time interval (like the 2016 seismic sequence). Additionally,
paleoseismic data inherently focus on surface-rupturing earthquakes, thus aliasing smaller seismic events, which however
could have caused significant damage.

Our study reinforces the need for detailed paleoseismic studies in a careful evaluation of the seismic hazard in such a densely
populated region. The study of capable faults affecting urbanized zones indeed provides valuable contexts that should inform
decision-making; we argue that conducting similar projects like ours could benefit other areas in and beyond the Italian
territory.

## Author contribution

Conceptualization and writing: FL, AMM, MFF, ES, VC, MCac, AMB. Field work, geological mapping, trenching and
logging: FL, AMM, MFF, ES, VC, MCac, AMB, PDM, FF, RG, PL, ASF, MP, KN, GT, FT, AP, FF, MG, FM, RNap,
RNav, RP, LMP, MR, AR, VR. Geophysical data acquisition and processing: LMP, VM, VR, VS, SU. Scientific discussion
and text revision: KN, AZ, GB, MCol, LG, PDM.

## Code/Data availability

All the data are publicly available in this work and in the Supplementary Material. The data presented are included in the
microzonation studies carried out in the municipalities of Central Italy affected by the seismic events starting from August
24, 2016, as provided by Ordinance No. 24 of May 12, 2017, issued by the Extraordinary Commissioner.



With Ordinance No. 55, Article 5, "Amendments to Ordinance No. 24 of May 12, 2017," the general criteria were approved for the use of Level 3 Seismic Microzonation studies in the reconstruction of the areas affected by the seismic events starting
from August 24, 2016. All the reports and data are publicly available at https://sisma2016data.it/microzonazione/ )in Italian (last accessed on the 29th, May 2025).

**Competing interests**

The Authors declare that they have no conflict of interest.

**Acknowledgements**

We are grateful to the owners of the land where we conducted geophysical prospecting and trench investigations. The assistance of the Rivodutri, Cantalice, Rieti and Cittaducale Municipalities is gratefully acknowledged. Special thanks to Matteo Carrozzoni, Geologist at "Commissario Straordinario Ricostruzione Sisma 2016", for the invaluable support to this project: without him our research would have been simply impossible. Thanks to Francesco Chiaretti and Domenico "Mimmo" Marchetti, authors of the local microzoning studies, for sharing their expertise and scientific discussion. We are
grateful to Enzo Sepe, INGV leader of the 2016 post-seismic capable fault project, indeed a scientific leader; and to Carlo Doglioni, INGV President, who strongly believed in the realization of this project.

**Financial support**

The agreements of scientific collaboration between INGV, ISPRA, and Università degli studi dell'Insubria "Ridefinizione delle Zone di Attenzione delle Faglie Attive e Capaci emerse dagli studi di microzonazione sismica effettuati nel territorio
comunale di Cittaducale (RI) e Rieti, interessati dagli eventi sismici verificatisi a far data dal 24 agosto 2016" signed on December 2020 and "Aggiornamento degli studi di microzonazione sismica a seguito degli approfondimenti dedicati alle zone delle faglie attive e capaci presenti nei territori dei Centri abitati di Rieti e Cittaducale (RI)" signed on April 2022 provided funding and the legal framework for the paleoseismic investigations in the Rieti Basin carried out by ISPRA and Università degli Studi dell'Insubria.
Research funded by European Union – NextGenerationEU – Mission 4 "Education and Research" – Component 2 "From Research to Business" – Investment 3.1 "Fund for the realization of an integrated system of research and innovation infrastructures" – Project IR0000037 – GeoSciences IR - CUP I53C22000800006
K. Nicoll acknowledges support and funding from the USA-Italy Fulbright Program.



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
