# Peer review of "Paleoseismic history of the intermountain Rieti Basin (Central 2 Apennines, Italy)"

_EGUsphere, 2025_

## Author Comment (AC1)

**Coulomb stress transfer preliminary results**

Rieti BASIN

Livio et al. (Submitted to NHESS)

**«Paleoseismic history of the intermountain Rieti Basin (Central Apennines, Italy)»**

**Model Building**

- Simplified fault traces
- Extended down to 10 km with a planar geometry
- Faults' attitude measured at surface.
- Max displacement constrained through paleoseismology and W&C94 empirical regression (i.e., 1 m for a Mw 6.6 eq).
- CSS is calculated by assuming a pure dip slip normal faulting on the northern and southern border fault.

**Slip on seismogenic fault:**
Max 1 m; tapering laterally.

For the MODEL 1 measured through paleoseismology (event D).

**Mechanical parameters:**
Poisson's Ratio:  : 0.25
Young's Modulus:  : 27000
Apparent Friction:  : 0.4

Free surface at zero elevation reference surface (no topography considered)

**MODEL 1: Max displacement to the north**

Southern border faults are mostly loaded (in EVENT D, those moved together with the main fault).

[Figure]

**MODEL 2: Max displacement at fault center**

The southern border faults are positively loaded at shallow depths. Northern border faults are not loaded.

[Figure]

**MODEL 3: Max displacement to the south**

Northern border faults are slighlty loaded and the eastern segment of the southern faults at depths > 2 km

---

## Author Response (AR1)

This document represents the response to the reviews of the paper entitled *"Paleoseismic history of*
*the intermountain Rieti Basin (Central Apennines, Italy)"*, which we submitted to the journal NHESS.

We wish to thank the reviewer Gerald Roberts for the positive feedback and taking the time to provide
improvements to our text. Here we provide a point-to-point answer to all the comments raised by the
reviewer. Original comments are shown in plain text, while answers are in *italic*.

The revised manuscript will be uploaded to the journal system.

**Reviewer's report on Livio et al. "Paleoseismic history of the intermountain Rieti Basin (Central**
**Apennines, Italy)" (Prof. Gerald Roberts - 1st July 2025)**

The manuscript presents new insights regarding the paleoseismic history of the Rieti Basin in Central
Italy, derived from the excavation of 17 paleoseismic trenches.

These results will be incredibly valuable, both locally for the seismic assessment of the Apennines,
but also in a general sense to aid the understanding of active faults worldwide. I was particularly
impressed by the number of trenches, the number of AMS dates, and the detailed trench logs and
photos. The descriptions of the trenches and the stratigraphy was also very detailed. The maps of the
sites were also of high quality. The observations are put into context of the Rieti basin faults in a clear
way. As such, I suggest accepting this manuscript with minor changes.

*Thanks for the very detailed review, below we address your comments and describe the changes*
*we will implement in the text.*

My comments as follows:

Overall, I was left wondering about how these observations fit in with the regional context of extension
across the Apennines. Would you like to add some text on this? For example, how does the timing and
magnitude of slip you have observed relate to that on other neighbouring faults? For example, were
other faults on the SW flank of the Apennines also slipping when the Rieti basin experienced fault slip?
I think it would be a missed opportunity not to say something about the regional context of the slip you
have documented.

*Thanks for this comment, it is certainly an important point and we missed this. We will add*
*some text about this in the discussion before the "disclaimer", near Line 502; as you suggest*
*below it is better to start the discussion with positive results. In fact, on nearby Quaternary*
*faults located near the SW flank of the Central Apennines we do have evidence of similar*
*surface faulting events that occurred in the same period as those observed in the Rieti basin*
*(e.g., Leonessa Fault: Mildon et al., 2022; Fiamignano Fault: Beck et al., 2018; Fucino Fault:*
*Gori et al., 2017; Liri Fault: Maceroni et al., 2022). In particular, slip histories recovered from*
*the 36Cl data using Bayesian MCMC modelling show that the Leonessa Fault, Fiamignano*
*Fault, Fucino Fault and Liri Fault exhibit a period of slip rate acceleration during the 7 to 5 kyr*
*BP and 2.5 to 1.5 Kyr BP time windows (Roberts et al., 2025). This is the same interval during*
*which the Rieti Basin faults generated the Earthquake Sequence 1 and 2 based on the*
*paleoseismic analyses described in the manuscript.*

*Refs to be added:*

*Beck, J., Wolfers, S. and Roberts, G.P., 2018, Bayesian earthquake dating and seismic hazard*
*assessment using chlorine-36 measurements (BED v1). Geoscientific Model Development,*
*11(11), pp. 4383-4397.*

*Gori, S., E. Falcucci, F. Galadini, M. Moro, M. Saroli and E. Ceccaroni, 2017, Geoarchaeology*
*and paleoseismology blend to define the Fucino active normal fault slip history, central Italy.*
*Quaternary International 451 (2017): 114-128.*

*Maceroni, D., Dixit Dominus, G., Gori, S., Falcucci, E., Galadini, F., Moro, M., & Saroli, M.*
*(2022). First evidence of the Late Pleistocene—Holocene activity of the Roveto Valley Fault*
*(Central Apennines, Italy). Frontiers in Earth Science, 10, 1018737.*

The text from lines 560 to 630 is well written and clear. However, reporting the ages in "BCE" means
readers need to convert the ages in years BP. Can you also add the ages as "years BP"? This would
make it much easier for most readers.

*Thank you for pointing this out. Considering that the recognized paleoearthquakes span over a*
*considerable time window and also historical times are covered, we choose to use the CE/BCE*
*date formatting. Nonetheless, we understand that, especially for older dates and for a direct*
*comparison with other results obtained in the Central Apennines, with different techniques,*
*we'll add the BP format in the text of Section 5.2 and in table 2 as well.*

Please try to provide the amount slip for each of the events, even if this is a minimum estimate. Your
approach is common amongst palaeoseismic trenching papers, where the magnitude of slip is
commonly challenging to interpret, and it is common for nothing to be reported (I know you do report it
for two of the events). However, it would be helpful to know what you think about the slip for all the
events. If you have no constraints on the slip (e.g. because dated material just fills a fissure) state this.
If you have some idea (e.g. colluvial wedge has a vertical extent of X cm), please state the value of X. If
you can identify piercing points please state the exact value of offset (e.g. vertical offset, and some
estimate of the fault dip with error bars). I feel it is better for you to say what you have observed about
possible offsets per event rather than someone trying to interpret this from your trench logs at a later
date – it is you who has the best chance of producing the best estimates, so it is a shame not to
provide this.

*Thanks for this note. Yes, it is indeed important to report all the possible available information*
*related to the identified paleoevents. We'll add our best estimate for the slip per event; even if*
*minimum. We'll clearly state if no estimation is available for specific causes. We'll add the*
*estimation in Table 2 and take care that in the trench description the constraints on the*
*estimated slip per event are clearly described.*

I also think that somewhere in your manuscript you need to define exactly what you mean by a
"cluster" (e.g. a spatial cluster? Or temporal cluster where the implied slip rate exceeds the long-term
slip-rate, or both? Or some other way of defining it?). You mention this word "cluster" on Line 634, yet
you do not define what you mean by this. Also, if these are temporal clusters, as implied by the text on lines 635-640 (e.g. Line 640 mentions a period of quiescence", implying the slip is clustered in time),
how do you define this? For example, how do you assess the relative weightings of the (i) average
recurrence interval over the long-term, (ii) the intervals defined by the preferred ages of the events
(e.g. the aperiodicity), whilst considering (iii) the uncertainties defined by the analytical uncertainties
on the 14C ages and the "brackets" they define? In other words, if you are going to talk about
clustering, you need to add some text about how you define clustering in your particular case.

*Thanks for pointing this out. Indeed the term "cluster" is misleading. We do not observe a*
*significant temporal clustering in the events. An analysis of the earthquake aperiodicity*
*suggests a moderately regular recurrence pattern for the events A-D (aperiodicity ca. 0.53).*
*After including also Events E and F the calculated aperiodicity is close to 1.12 (i.e., a highly*
*irregular recurrence pattern). Nonetheless, the stratigraphic record for the 20 - 10 ka interval*
*can be considerably incomplete.*

*After these considerations, to avoid confusion, we substituted the term "cluster" with*
*sequences. We here intend a sequence as a series of characteristic earthquakes rupturing*
*adjacent fault segments or adjacent faults over a short time period. So, it is mainly referring to*
*a spatial criterion rather than a temporal one. We further explore the possible spatial*
*interaction among the basin-bounding faults by running some Coulomb stress transfer models*
*(for full details, see below the answer to the comment at line 655).*

Line 502 – It might be better to start the discussion with a paragraph that states the main findings, and
then go on to set the scene for the coming discussion by outlining the main points that will be covered.
The reader will then know what is to come next. Starting with disclaimers is a little dull and off-putting.
Instead start with positive outcomes.

*Yes, we agree. We'll change the text accordingly (see the above response to the main*
*comments).*

Line 521 – you say it is "impossible" to find suitable sites, but what you really mean is that it is
challenging and beyond your current capabilities. Please change the text.

*Agreed. Text changed to "challenging".*

Line 535 – The text "underwent slope recession during the Last Glacial, are now very limited" is very
vague. Imagine the reader who is not familiar with the Apennines; will they understand what this
means? Re-phrase to describe the precise interplay between tectonic slip-rates and the change in
erosion rate that occurred at the end of the last glacial maximum, perhaps even stating values in
mm/yr, because such values are available in the literature (e.g. Tucker et al. 2011, JGR), and it is your
duty to point this out to readers who do not know the Apennines.

*We are grateful to the reviewer for this comment. Indeed, the interplay between tectonic slip-*
*rates and erosion rates in the Apennines of Central Italy, particularly on carbonate bedrock*
*fault scarps, is a well-studied example of how climatic and tectonic processes interact on*
*geologic timescales. At the end of the Last Glacial Maximum (LGM), significant changes in*

*surface processes—especially erosion—impacted how fault scarps are preserved and*
*measured, with implications for estimating long-term slip rates.*

*During the last glaciation, the high elevations of the Central Apennines (reaching up to 2900 m)*
*hosted mountain valley glaciers, as evidenced by the presence of moraines and other glacial*
*landforms. In areas beyond the extent of glaciation, periglacial conditions prevailed. On*
*mountain slopes carved in pure carbonate bedrock, rapid erosion and sedimentation, at rates*
*higher than 0.2–0.4 mm/year (Tucker et al., 2011), led to the formation of alluvial fans emerging*
*from ice-free mountain valleys and slopes. These rates exceeded the typical fault throw rates*
*(e.g., Roberts and Michetti, 2004), as demonstrated by fan surfaces and colluvial slopes on*
*fault hanging walls that are graded to the adjacent footwall bedrock slopes.*

*As the glaciers retreated, the reestablishment of temperate vegetation helped stabilize both*
*the alluvial fans and surrounding slopes, while stream discharges declined. This transition*
*resulted in the smooth hillsides that characterize landscapes shaped by former periglacial*
*activity. Fan surfaces, bedrock slopes, and moraines are often covered by a thin*
*(approximately 0.5–1.0 m) layer of soil enriched in organic material and, in some places,*
*volcanic components—deposited during and after glacial retreat.*

*Elsewhere, the end of glaciation is marked by frontal moraines overlain by fluvial outwash or*
*deposits from meltwater lakes. These sediments often contain palaeovegetation and volcanic*
*ash from nearby eruptions, providing material suitable for radiocarbon dating and*
*tephrochronology. A large dataset of such dates allows for the determination of both absolute*
*and relative ages of glacial and periglacial features, and facilitates correlations with climate*
*records from Tyrrhenian Sea cores and other oceanic and continental archives. The final major*
*phase of glacial retreat occurred around 18–16 thousand years ago, coinciding with a*
*significant shift in $\delta^{18}O$ values observed in Tyrrhenian Sea cores and other marine records,*
*confirming a major climatic transition.*

*Presently, normal fault scarps carved in pure carbonate bedrock cut through these glaciation-*
*related landscapes. In many locations, these scarps expose Mesozoic carbonate platform*
*bedrock in their footwalls and exhibit minimal degradation.*

*However, marly-limestone bedrock from the Meso-Cenozoic Umbria-Marche-Sabina pelagic*
*environment of the Central Apennines, where the Rieti basin belong, behave differently from*
*pure carbonate platform bedrock of the Abruzzi Mesozoic facies. In our study we mainly*
*investigated bedrock fault scarps carved in the marly limestone of the Scaglia Formation.*
*Modern erosion rates on the Scaglia Formation are higher than those observed on the Abruzzi*
*carbonate platform (0.016 ± 0.005 mm/yr; Tucker et al., 2011). In fact, prominent bedrock fault*
*scarps in the Umbria-Marche-Sabina region (for instance in the Rieti, Leonessa and Norcia*
*basins) are only visible where pure carbonate formations such as Calcare Massiccio and*
*Maiolica are outcropping. This point is relevant and was already discussed by Blumetti et al.*
*1993.*

*We will add this discussion at the beginning of par 5.1.*

*Refs to be added:*

*Tucker, G. E., McCoy, S. W., Whittaker, A. C., Roberts, G. P., Lancaster, S. T., & Phillips, R.*
*(2011). Geomorphic significance of postglacial bedrock scarps on normal-fault footwalls.*
*Journal of Geophysical Research: Earth Surface, 116, F01022, 1-14,*
*doi:10.1029/2010JF001861, 2011*

Line 545 – you imply that studying scarps is challenging due to human effects, but then show you can
get results. So, don't you think you are overstating the effect of humans on ruining scarps? This will be
confusing and possibly misleading for readers unfamiliar with the Apennines. If one does a good job of
picking sites one can avoid human effects. This is nothing new and you show it very well with your
work. Even in sites with less forest and a lower value for the population density through time one still
has to work hard to find suitable sites. So, please clarify on this point. We don't want people being
put-off from working on the faults in the Apennines because they gain the false impression that it is
impossible to get results due to human degradation of the sites.

*We thank the reviewer for this comment. We agree that if it is possible to select trenching sites*
*not influenced by human activity this problem can be avoided. However, due to the logistic*
*constraints of our study that was primarily focused on capable fault detailed mapping for post-*
*emergency reconstruction microzoning, we were forced to trench fault scarps clearly affected*
*by human effects. We will make this clearer in the revised text. We think we are not overstating*
*the anthropic effects on scarp morphology preservation. We just emphasize the need for*
*taking properly into account human impact on fault scarp erosion and sedimentation at sites*
*clearly modified by agriculture and human land use.*

Line 555 – Please state some values in mm/yr for slip-rates on the faults you mention to prove your
claim that they are "similar".

*Ok, we will add published values of postglacial slip-rates for the Monte Morrone Fault (0.20-*
*0.40 mm/yr; Puliti et al., 2024), Paganica Fault (0.25-0.30 mm/yr; Cinti et al., 2011), Montereale*
*Fault (0.30-0.40 mm/yr; Cinti et al., 2018), and Monte Vettore Fault (minimum 0.26-0.38*
*mm/yr; Cinti et al., 2019). For the Mt. Vettore Fault more recent slip rates estimates are 0.8-1*
*mm/yr (Puliti et al 2020); 0.7-1.2 mm/yr (based on 36Cl exposure dating of bedrock fault plane;*
*Pousse Beltran et al. 2022); ranging from a minimum of 0.4 to a maximum of 1.3 mm/yr (Galli et*
*al., 2019).*

*References:*

*Puliti, I., Pizzi, A., Benedetti, L., Di Domenica, A., & Fleury, J. (2020). Comparing slip distribution*
*of an active fault system at various timescales: Insights for the evolution of the Mt. Vettore-Mt.*
*Bove fault system in Central Apennines. Tectonics, 39(9), e2020TC006200.*

*Pousse-Beltran, L., Benedetti, L., Fleury, J., Boncio, P., Guillou, V., Pace, B., ... & Aster Team.*
*(2022). 36Cl exposure dating of glacial features to constrain the slip rate along the Mt. Vettore*
*Fault (Central Apennines, Italy). Geomorphology, 412, 108302.*

*Galli, P., Galderisi, A., Peronace, E., Giaccio, B., Hajdas, I., Messina, P., ... & Polpetta, F.*
*(2019). The awakening of the dormant Mount Vettore fault (2016 central Italy earthquake, Mw*
*6.6): Paleoseismic clues on its millennial silences. Tectonics, 38(2), 687-705.*

Line 565 and Figure 15 – It is not very easy to move between the "black-boxes" on Figure 15 and the
events and ages on Table 2. Please improve this. In fact, I found Figure 15a quite poor and hard to
understand. I had to make my own figure to try to understand the timings of events using the data in
the table. I don't think the "fading colours" are clear in their meaning, e.g. purple to white and dark-
green to white, as you have not defined what "white" means. Please use solid symbols (something like
error bars on a cross-plot?). I also think the "red" blocks showing the 14C dates are unclear in their
meaning, so please use a symbol with the analytical error bars and the calibrated age as a data point
(so a data point and error bars). Overall, this Figure is your key result and its present design makes it
hard for the reader to extract the results – as I say I had to try to make my own figure. Please improve
this because this is what will improve the citation of your paper. If people don't understand the figures
and hence the paper they will not cite it.

....

Figure 15b is good for seeing the spatial pattern of clustering, but the text on line 635 implies a pattern
of temporal clustering, which I feel is not adequately displayed on Figure 15b. Can you provide a figure
that shows the temporal pattern of clustering?

*Thanks for pointing this out. Yes, Figure 15a needs to be definitely improved. We'll follow your*
*suggestions by using clearer graphical symbology in a simpler chart. Also, we'll improve the*
*graphical rendering of the timeline with the proposed earthquakes. We agree that this figure*
*could be the real core of the paper, it deserves to be improved.*

Line 650 – I agree they look like release faults. However, you don't need structural inheritance to form
these. If there is evidence for pre-existing structures in this specific example, please tell us what the
evidence is. If there is no evidence, don't claim this.

*Inheritance is not needed to develop release faults, as you are correctly stating. On the*
*contrary, release faults are typically much shallower than the main basin-bounding fault and*
*their role is expected to be less and less necessary at depth. Inheritance has been invoked here*
*for the consideration that in the Central Apennines faults can be segmented into short faults*
*with considerable slip gradients along strike. Indeed, the recent work by Caportorti and Muraro*
*(2025) showed several examples of Quaternary faults that re-activated at depth previous*
*Miocene normal faults and, at surface, display a fault trace close to coincident to the Miocene*
*ones (e.g., the Norcia, Mt. Boragine, Leonessa and Mt. Marine faults – see Capotorti and Muraro*
*and Supplementary Material of their work for a comparison). The faults bounding the Rieti Basin*
*are located close to inherited rift-related faults, as well. To the west, the basin bounding fault*
*runs almost along the inherited Sabina Paleofault (Galluzzo and Santantonio, 2002). To the*
*east, the main basin-bounding fault is running at the margin of a series of aligned mesozoic*
*structural highs that (i.e., the so-called pelagic Carbonate Platforms - PCP; specifically, the*
*Lisciano, Mt. Rosato and Polino ones; Capotorti and Muraro, 2024). These have already been*

*interpreted as related to a secondary structural high in the hanging wall of the Sabina paleofault*
*(see Galluzzo and Santantonio, 2002, their Figure 30). We'll add this specific explanation to the*
*text to make it clearer.*

Line 655 – Have you conducted a stress transfer calculation, or are you speculating? It might be worth
doing a stress transfer calculation to make sure you are correct. After all, the value of stress on the
receiver fault is influenced by the dip and strike of the fault, and the depth considered, as well as its
position relative to the ruptured fault.

*Some models have been run to test the hypothesis. Under very simple assumptions and*
*boundary conditions, given a simple geometry of the modeled faults, we observed that it is the*
*along-strike slip distribution on the eastern border fault that primarily determines the CSS on*
*the receiving faults. In the attached pdf file you can check the results from the CSS modeling.*
*Northern and southern boundary faults can be slightly loaded in the shallowest sectors, by the*
*movement of the eastern border fault. In particular, the simulation of a slip event like the D*
*earthquake we uncovered through paleoseismology is slightly charging the southern border*
*fault that, consistently, showed evidence for a contemporary movement also by means of a*
*paleoseismological approach.*

*We do not want to discuss in detail the CSS modeling in this work, rather, we prefer to*
*investigate these results in a future companion paper addressed specifically to this issue.*

Line 658 – You say they are not compatible with a single phase of extension, but they are compatible if
you believe they are release faults (which you say a few lines earlier). I suggest removing the "hardly
compatible" phrase as it contradicts your earlier claim of release faults (also this is incorrect English –
it is either compatible or incompatible – "hardly compatible" is not correct English).

*Thanks for pointing this out. Yes, indeed we were referring to a simple Andersonian model of*
*faulting under tectonics. Nonetheless, we'll change it as suggested.*

Line 660 – why not cite a paper on the 1983 Borah peak ruptures where ruptures fanned out at the NW
end onto the Willow Creek Hills at a relatively high angle to the main fault. This supports what you
claim.

*Thanks for the suggestion. Yes, indeed we'll add the Borah Peak 1983 earthquake as an*
*additional example (Figure 16f).*

Line 654 – change the word "descent" - incorrect English. "originate"?

*Text will be changed as suggested.*

On Figures where you show the "vertical offset", e.g. Figure 3, please state what feature is offset. I
presume it is the topographic slope, so I also presume that this is the vertical offset across the topographic scarp formed after the slopes stabilised. However, I don't think you have stated this, so I
am not sure. Would you like to comment on how the magnitude of slip in the trenches relates to the
"vertical offset" on the profile?

*Thanks to Gerald Roberts for this detailed comment. The offset feature is the topographic*
*slope in the scarp footwall, as presumed by the reviewer. This footwall slope is carved in the*
*marly limestone Scaglia Bianca Formation; the fault juxtaposes Scaglia Bianca Formation*
*against the Villafranchian lacustrine clay and silt deposits. The trench site is on the Southern*
*border of the Apoleggia village. Due to agricultural impact, the sedimentation of slope deposits*
*at this site was quite relevant during historical times, as demonstrated by the age of colluvial*
*deposits in the fault hanging wall. Therefore, in Figure 3, the 5.2 m vertical offset is calculated*
*by projecting the topographic slope in the footwall below the thickness of historical slope*
*deposits. In fact, ERT profiles clearly show the position of the top of the Pleistocene lacustrine*
*deposits, which is parallel to the footwall slope and can be regarded as a proxy for the glacial*
*topographic slope. Therefore, at this site the scarp height is lower than the vertical postglacial*
*fault offset.*

*We follow the same approach also for the Cantalice scarp, where the footwall bedrock is in the*
*same Scaglia Bianca Formation, and the hanging wall is in the Villafranchian lacustrine*
*deposits; the human impact on slope deposits is similar, based on dating of hanging wall*
*samples collected in the trenches; and ERT profiles show a similar geometry for the top of*
*Pleistocene lacustrine deposits and glacial slope. We will add this text to the discussion.*

On Figures the photos are in places "greyed-out" and covered by an interpretation. Is it possible to
provide the un-interpreted photos as well, perhaps in an Appendix?

*Thanks for the suggestion. Yes, We'll provide an uninterpreted photomosaic of each trench wall*
*in the Supplementary Material.*

On Figure 7 please change the text "unit is truncated, to the top, by an erosive surface – Bedrock" That
is obviously incorrect English, and does not state the field relationship precisely or correctly. It is not
"to the top".

*Thanks for pointing this out. We'll correct the text.*

On Figure 9, there is mention of slickensides on unit 12 – do you have any measurements of the slip
vector to show if these faults are dip-slip or oblique slip?

*In this case, slickensides are a characteristic of vertisols, typical of clay soils subject to mass*
*movements or vertisification phenomena. They are not indicators of kinematics.*

The graphics on the inset ERT plot for Figure 11 are poorly quality (pixellated) – can you improve the
resolution of this figure?

*We are sorry, but we cannot further improve resolution of the enlargement of the ERT profile in*
*figure 11. We underline that in any case this is only a graphical limitation and is not changing, in*
*any way, the resulting interpretation.*

Figure 13 is hard to visualize the context of the image. Can you provide a context photo to show the
aspect of the slope?

*Thanks for the suggestion. We have updated the image by adding a new photo showing the*
*local morphological setting.*

Overall, I enjoyed the paper and found it quite inspiring. Well done.

Professor Gerald Roberts, 1st July 2025

**RC2**: 'Comment on egusphere-2025-2531', Gordon Woo, 14 Jul 2025

This is a very detailed study which plugs a number of significant gaps in knowledge.  To their credit, the
authors have themselves identified limitations of their work.  One of the most crucial seismic hazard
parameters is the maximum magnitude. The authors conclude as follows:

Our results indicate that the maximum credible earthquake in the Rieti Basin is in the order of
magnitude Mw 6.5, which is consistent with the general setting of the Central Apennines. Given the
resolution of chronological constraints obtainable with radiocarbon dating techniques in
paleoseismic trenches, we cannot disentangle the occurrence of a single earthquake as compared to
multiple earthquakes occurring over a short time interval (like the 2016 seismic sequence).
Additionally, paleoseismic data inherently focus on surface-rupturing earthquakes, thus aliasing
smaller seismic events, which however could have caused significant damage.

Given the ambiguity over the release of seismic energy in one large event, or a sequence of lesser
events, it would be helpful if the authors could provide more substantive discussion over the
maximum magnitude in the Rieti Basin. In particular, could the maximum magnitude be as high as 6.7
or 6.8?

*We wish to thank the reviewer Gordon Woo for the positive feedback and taking the time to*
*provide improvements to our text. Here we provide a point-to-point answer to all the*
*comments raised by the reviewer. Original comments are shown in plain text, while answers*
*are in italic.*

*The revised manuscript will be uploaded to the journal system.*

*We thank the referee for the useful comment. We did not discuss in detail the maximum*
*earthquake magnitude estimate, so in the manuscript we will expand the topic, according to*
*the following reasoning. We derive constraints on Maximum Magnitude in the Rieti Basin from*
*historical seismicity and earthquake surface rupture length. Epicentral intensity data on*
*historical earthquakes, such as those occurred on 76 BCE and 1298 CE, show Io = X MCS*

*(Mercalli Cancani Sieberg scale) in the Italian seismic catalogue. Using Magnitude Vs. Intensity*
*correlations this gives M6.40 for the 76 BCE earthquake (Guidoboni et al. 2018, 2019), and*
*M6.26 for the 1298 earthquake (Brunamonte et al., 1993; Rovida et al., 2022). Data on similar*
*ancient historical events is obviously not very detailed. However, they provide a reasonable*
*minimum value.*

*As for estimates based on earthquake surface rupture length, based on our extensive coverage*
*of paleoseismic trench sites, we assume that rupture of the full 21 km length of the Rieti Basin*
*eastern master fault is a credible hypothesis. Using literature empirical relations (e.g., Wells*
*and Coppersmith, 1994; Pavlides and Caputo, 2004) this gives a value of ca. Mw 6.5. Since*
*standard deviations of published empirical relations are ca. 0.3, we cannot rule out a Mw max*
*of 6.8. However, comparison with well-studied recent earthquakes in the Central Apennines*
*indicates that Mw 6.5 is a reasonable estimate. We believe that uncertainty in available data*
*does not allow a more detailed numerical analysis, which is beyond the scope of the*
*manuscript.*

*References:*

*Pavlides, S., and Caputo, R.: Magnitude versus faults' surface parameters: quantitative*
*relationships from the Aegean Region, Tectonophysics, 380(3-4), 159-188, 2004.*

*Wells, D. L., and Coppersmith, K. J.: New empirical relationships among magnitude, rupture*
*length, rupture width, rupture area, and surface displacement. Bulletin of the seismological*
*Society of America, 84(4), 974-1002, 1994.*

**RC3**: 'Comment on egusphere-2025-2531', Anonymous Referee #3, 15 Jul 2025

This manuscript presents a thorough paleoseismological investigation of the Rieti Basin in the Central
Apennines. Through an extensive trenching involving 17 excavation sites along active fault segments,
the authors successfully identified 15 paleoearthquakes during the last ca. 20 kyr. The scope and
resolution of the work are impressive, demonstrating a significant investment in fieldwork,
stratigraphic analysis, and chronological interpretation. While the study is of high quality, addressing
a few specific issues could further enhance its clarity and impact.

*We wish to thank the reviewer for the positive feedback and taking the time to provide*
*improvements to our text. Here we provide a point-to-point answer to all the comments*
*raised by the reviewer. Original comments are shown in plain text, while answers are in*
*italic.*

*The revised manuscript will be uploaded to the journal system.*

1. Figure 2 contains only three subfigures, but the caption refers to a subfigure "d)", which
appears to be an error and should be corrected. In addition, the caption mentions a star
symbol, but it is not visible in Figure 2a; the authors should ensure that all referenced symbols
are clearly displayed. Lastly, in Figure 2b, the GPR17 and GPR18 survey lines are difficult to
distinguish.

*Thanks for spotting this error. We'll amend the caption as follows:*

   *"Figure 2: Study Area along the Northern Border fault: a) simplified geological map; b)*

   *and c) panels display detailed views on the two studied Sectors with the traces of the*

   *capable faults, of the geophysical investigations and the footprints of the excavated*

   *paleoseismological trenches; red dot in Fig. 2a is the Piedicolle trench Site from*

   *Michetti et al. (1995)."*

2. On page 13, line 220, Figure 3d does not include the ERT_14 section. Therefore, the sentence

"Figure 3d shows the ERT_14 section (see location in Fig. 2b);" should be corrected to

"Figure 3 shows the ERT_14 section (see location in Fig. 2b);".

   *Thanks for spotting this error. We'll change it accordingly.*

3. Please consider indicating the fault in Figure 3d (ERT_15 result) to help clarify the relationship between resistivity anomalies and fault structures. Additionally, the main text does not provide a detailed interpretation of this figure. The absolute error associated with the Wenner–

Schlumberger array is smaller than that of the dipole–dipole array (Fig. 13d). Could this be interpreted as an indication that the Wenner–Schlumberger result is more reliable, or might it instead suggest that the inversion result is overfitting the observed data?

   *Yes, we'll add the fault interpretation on Figure 3d. As for the errors in the two arrays: in*

   *comparative studies, Wenner-Schlumberger arrays have shown better stability and*

   *lower RMS errors during inversion, particularly in environments with moderate to low*

   *resistivity contrasts. Dipole –Dipole is better performing in direct imaging of possible*

   *fault planes, even if noisier. Nonetheless, both the inversions show really good*

   *statistical fitting after inversion. Additionally, the software we used for ERT inversion*

   *(REs2dInv) offers a robust inversion method that minimizes absolute rather than*

   *squared differences. This is less sensitive to outliers and helps prevent overfitting to*

   *noisy data.*

4. On page 16, "CAM_ERT_02" should be corrected to "CAM_ERT_17" to ensure consistency with the context.

   *Agreed and modified accordingly.*

5. The fault zone in ERT17 (Figure 4) shows a weaker resistivity contrast than in ERT14 (Figure 3).

It would be helpful if the authors could clarify whether this difference arises from subsurface lithological variations, differences in survey configuration, or other influencing factors, as this would offer useful guidance for ERT applications in fault zone imaging.

   *Thanks for pointing this out. The two ERT profiles are reaching different lithologies. In*

   *ERT 14 bedrock is outcropping just next to the ERT line, and the very high resistivity*

   *values are referred directly to a fractured bedrock. In Section ERT17, instead, the*

   *Villafranchian succession is imaged (i.e., pedogenized fine sands and silts). Possibly,*

   *bedrock is imaged only in the lowermost part of the section but due to the very high*

*uncertainty characterizing that sector of the tomography, we avoided giving direct*
*interpretations.*